



# A systematic examination of the relationships between CDOM and DOC in inland waters in China

Kaishan Song[1], Ying Zhao[1,2], Zhidan Wen[1], Chong Fang[1,2], Yingxin Shang[1]

[1]Northeast Institute of Geography and Agroecology, CAS, Changchun, 130102, China

[2] University of Chinese Academy of Sciences, Beijing 100049, China

Corresponding author's E-mail: songks@iga.ac.cn; Tel: 86-431-85542364

**Abstract:** Chromophoric dissolved organic matter (CDOM) plays a vital role in the biogeochemical cycle in aquatic ecosystems. The relationship between CDOM and dissolved organic carbon (DOC) has been investigated, and the significant relationship lays the foundation for the estimation of DOC using remotely sensed imagery data. An algorithm has been developed to retrieve DOC via CDOM absorption ($a_{CDOM}$) at 275 and 295 nm for coastal waters, but it is still unclear for the relationship between DOC and $a_{CDOM}$ in other types of waters. The current study examined the samples from freshwater lakes, saline lakes, rivers and streams, urban water bodies, and ice-covered lakes in China. The regression model slopes for DOC versus $a_{CDOM}$(275) ranged from extreme low 0.33 (highly saline lakes) to 1.03 (urban waters) and 3.13 (river waters). The low values were observed in saline lake waters and waters from semi-arid or arid regions where strong photo-bleaching is expected due to thin ozone layers, less cloud cover, longer water residence time and daylight hours. In contrast, high values were found in waters developed in wetlands or forest in Northeast China, where massive organic matter was transported from catchment to





waters. The study also demonstrated that stronger relationships between CDOM and
DOC were revealed when $a_{CDOM}(275)$ were sorted by the ratio of $a_{CDOM}(250)$ to
$a_{CDOM}(365)$, which is a tracer for the CDOM absorption with respect to its
composition, and the determination of coefficient of the regression models ranged
from 0.78 to 0.99 for different groups of waters. Our results indicated the
relationships between CDOM and DOC are variable for different inland waters, and
therefore remote sensing models for DOC estimation through linking with CDOM
absorption need to be tailored according to water types.
**Keywords:** Absorption, CDOM, DOC, regression slope, saline water, fresh water





## 1. Introduction

Compared with other terrestrial ecosystems, e.g., forest and grassland, inland waters
only occupy a small fraction (3.5%) of the earth surface (Verpoorter et al., 2014).
However, they play a disproportional role for the global carbon cycling with respect
to carbon transportation, transformation and carbon storage (Tranvik et al., 2009;
Verpoorter et al., 2014; Yang et al., 2015). According to Tranvik et al. (2009), 2.9 Pg
C was transported from terrestrial ecosystems to inland waters every year, of which
about 0.6 Pg C was buried in the lake sediment, 1.4 Pg C was released into the air as
$CO_2$ or $CH_4$, and the rest of 0.9 Pg C was exported to the ocean via river channels.
However, the amount of dissolved organic carbon (DOC) stored in the inland waters
is still unclear or the uncertainty is still needed to be evaluated (Tranvik et al., 2009).
Determination DOC concentration is straightforward through field sampling and
laboratory analysis (Findlay and Sinsabaugh, 2003). However, there are millions of
lakes in the world, and many of them are remote and inaccessible, making it
impossible to evaluate DOC concentration using routine approach (Cardille et al.,
2013; Brezonik et al., 2015). Researchers have found that remote sensing might
provide a promising tool for quantification of DOC of inland waters at large scale
through linking DOC with chromophoric dissolved organic matter (CDOM),
particularly for these inland waters situating in remote region with less accessibility
(Cole et al., 2007; Tranvik et al., 2009; Kutser et al., 2015; Brezonik et al., 2015).
CDOM is one of the largest bioactive reservoirs of organic matter on the earth
(Para et al., 2010), influencing light transmittance in aquatic ecosystems (Vodacek et




al., 1997; Williamson and Rose, 2010). As one of the optically active constituents
(OACs) in waters, CDOM can be estimated through remotely sensed signals (Yu et al.,
2010; Kutser et al., 2015), and is acted as a proxy in many regions for the amount of
DOC in the water column. As shown in Fig.1, CDOM and DOC in the aquatic
ecosystems are mainly originated from external (allochthonous) and internal
(autochthonous) sources, in addition to directly discharge from anthropogenic
activities (Zhou et al., 2016). Generally, the autochthonous CDOM is essentially
originated from algae and macrophytes, and mainly consists of various compounds of
low molecular weights (Findlay and Sinsabaugh, 2003; Zhang et al., 2009). While, the
allochthonous CDOM is mainly derived from the surrounding terrestrial ecosystems,
and it comprises a continuum of small organic molecules to highly polymeric humic
substances with compounds typically ranging from 100 to 100,000 Da. In terms of
CDOM originates from anthropogenic, it contains fatty acid, amino acid and sugar,
thus the composition of CDOM is more complex than that from natural systems
(Zhou et al., 2016; Zhao et al., 2016). Hydrological factor also affects the DOC and
CDOM characteristic. The concentrations of and the relationship between CDOM and
DOC in river waters depend on many factors, in which the water type, the seasonality
and climatology, the typology of the water, the surrounding landscapes. Particularly,
the discharge and catchment area are the most important ones (Neff et al., 2006;
Spencer et al., 2012; Alvarez-Cobelas et al., 2012).

**[Insert Fig.1 about here]**

CDOM is a major light-absorbing substance, which is responsible for much of the



Hydrology and
color in waters (Reche et al., 1999). The chemical structure and origin of CDOM can
be characterized by its absorption coefficients ($a_{CDOM}(\lambda)$) and spectral slopes (De
Haan and De Boer, 1987; Helms et al., 2008). Weishaar et al. (2003) has proven that
the carbon specific absorption coefficient at 254 nm, e.g., $SUVA_{254}$ is a good tracer
for the aromaticity of humic acid in CDOM, while the ratio of CDOM absorption at
250 to 365 nm ($a_{CDOM}(250/365)$, herein, M values) has been successfully used to track
the changes in DOM molecule weight (De Haan and De Boer, 1987; Zhang et al.,
2010) and absorption intensity (Song et al., 2013). Biodegradation and
photodegradation are the major processes to determine the transformation and
composition of CDOM (Findlay and Sinsabaugh, 2003). With prolonged sunlight
absorbed by CDOM, some of the colored fraction is lost by the photobleaching
processes (Miller et al., 1995; Zhang et al., 2010), which can be measured by the light
absorbance decreasing at some specific (diagnostic) wavelength, e.g., 250, 254, 275,
295, 365 or 440 nm. It should be noted that $a_{CDOM}(440)$ is usually used by remote
sensing community due to this wavelength is less affected by phytoplankton (Lee et
al., 2002). Under this circumstance, the relationship between CDOM and DOC varies
since CDOM loses color while the variation of DOC concentration is almost
negligible. Saline or brackish lakes in the arid or semi-arid regions generally expose
to longer sunlight radiation, thus CDOM absorbance decreases, while DOC is
accumulated due to the longer residence time (Curtis et al., 1997; Song et al., 2013;
Wen et al., 2016). Compared to photodegradation on CDOM, the biodegradation
processes by microbes are much complicated, and extracellular enzymes are the key





substance required to decompose the high-molecular-weight CDOM into
low-molecular-weight substrates (Findlay and Sinsabaugh, 2003; Romera-Castillo et
al., 2012). With compositional change, the absorption feature of CDOM and its
relation to DOC varies correspondingly, but the relationship between CDOM and
DOC is far from solved (Gonnelli et al., 2013). In addition, the $SUVA_{254}$ and
$a_{CDOM}(250/365)$ may be used to classify CDOM into different groups and enhance the
relationship with DOC based on CDOM absorption grouping.

Some studies have researched the spatial and seasonal variations of CDOM and

DOC in ice free season in lakes, rivers and oceans (Vodacek et al., 1997; Neff et al.,
2006; Stedmon et al., 2011; Brezonik et al., 2015), but less is known about saline
lakes (Song et al., 2013; Wen et al., 2016), particularly urban waters influenced by
sewage effluent and waters with ice cover in winter (Belzile et al., 2000, 2002; Zhao
et alb., 2016). The relationship between DOC and CDOM lays the foundation for the
remote sensing estimation of DOC in both inland waters (Yu et al., 2010; Griffin et al.,
2011; Zhu et al., 2014; Brezonik et al., 2015) and marine (Hoge et al., 1996; Bricaud
et al., 2012; Nelson et al., 2012). The significant relationship between CDOM and
DOC was observed in the Gulf of Mexico, and stable regression model was
established between DOC and $a_{CDOM}(275)$ and $a_{CDOM}(295)$ (Fichot and Benner 2011).
Similar results were also found in other estuaries along a salinity gradient, for
example the Finish Gulf (Kowalczuk et al., 2006) and the Chesapeake Bay (Le et al.,
2013). However, Chen et al. (2004) found that the relationship between CDOM and
DOC was not conservative due to estuarine mixing or photo-degradation. Similar





arguments were raised for Congo River (Spencer et al. 2009) and waters across
mainland USA (Spencer et al., 2012). The study on the relationship between DOC and
CDOM in Lake Taihu found a relatively stable relationship for water samples
collected in different seasons except winter (Jiang et al. 2012). However, seasonal
variations were observed in some studies due to the mixing of various endmembers of
CDOM from different terrestrial ecosystems and internal source (Zhang et al., 2010;
Spencer et al., 2012; Zhou et al., 2016). Along with laboratory measurements,
portable instruments deployed in river or streams provide great potential to quantify
DOC and CDOM at very dynamic manner (Lee et al., 2015; Yu et al., 2016).
According to Fig.1, the proposed hypothesis suggests that the main source of
CDOM and DOC in different waters vary, coupled with biogeochemical processes
(photobleaching and microbial degradation), resulting in the compositional
differences, and ultimately affects CDOM absorption and its relationship with DOC.
Hydrological feature and anthropogenic processes further cause the relationship
between CDOM and DOC varies both in time and space. Remote sensing technology
has increasingly played a vital role in quantifying carbon cycling in inland waters
(Tranvik et al., 2009; Raymond et al., 2013). However, the prerequisite is to
systematically examine the relationship between CDOM and DOC. In this study, the
characteristics of DOC and CDOM in different inland waters across China were
examined to determine the spatial feature associated with landscape variations,
hydrologic conditions and saline gradients. The objectives of this study are to: 1)
examine the relationship between CDOM and DOC concentrations across a wide



range of waters with various physical, chemical and biological conditions, and 2)
develop a model for the relationship between DOC and CDOM based on the sorted
CDOM absorption feature, e.g., the ratio of $a_{CDOM}(250/365)$ with aiming to improve
the regression modeling accuracy.

## 2. Materials and Methods

The dataset is composed of five subsets of samples collected from various types of
waters across China (Table 1, Fig.2), which encompassed a wide range of DOC and
CDOM. The first dataset (n = 288; from early spring 2009 to late October 2014)
includes samples collected in freshwater lakes and reservoirs during the growing
season with various landscape types. The second dataset (n = 345; from early spring
2010 to late mid-September 2014) includes samples collected in brackish to saline
water bodies. The third dataset (n =322; from early May 2012 to late October 2014)
includes samples collected in rivers and streams across different basins in China. In
addition, 69 samples were collected from three sections along the Songhua Rive, the
Yalu and the Hunjiang River during the ice free period in 2015 to examine the impact
of river flow on the relationship between DOC and CDOM (see Fig.S1 for location).
The fourth dataset (n = 328; from 2011 to 2014 in the ice frozen season) includes
samples collected in Northeast China in winter from both lake ice and underlying
waters. The fifth dataset (n = 221; from early May 2013 to mid-October 2014)
collects samples in urban water bodies, including lakes, ponds, rivers and streams,
which were severely polluted by sewage effluents. City maps and Landat imagery
data acquired in 2014 or 2015 were used to delineate urban boundaries with ArcGIS





10.0 (ESRI Inc., Redlands, California, USA), and water bodies in these investigated
cities constrained by urban boundaries were considered as urban water bodies. Except
river samples, the sampling dates, water body names and locations of other types of
water bodies were provided in supplementary Table S1-3.
**[Insert Fig.2 about here]**
**2.1 Water quality determination**
Water samples were collected approximately 0.5m below the water surface at each
station, generally locating in the middle of water bodies. Water samples were
collected in two 1 L amber HDPE bottles, and kept in coolers with ice packs in the
field and kept in refrigerator at 4℃ after shipping back to the laboratory; all samples
were preprocessed (e.g., filtration, pH and electrical conductivity (EC) determination)
within two days in the laboratory. Water salinity was measured using DDS-307 EC
meter (μS/cm) at room temperature (20±2℃) and converted to *in situ* salinity units
(PSU) in the laboratory. Water samples were filtered using Whatman cellulose acetone
filter with pore size of 0.45 μm. Chlorophyll-a (Chl-a) was extracted and
concentration was measured using a Shimadzu UV-2050PC spectrophotometer (Song
et al., 2013). Total suspended matter (TSM) was determined gravimetrically using
pre-combusted Whatman GF/F filters with 0.7μm pore size, details can be found in
Song et al. (2013). DOC concentrations were measured by high temperature
combustion (HTC) with water samples filtered through 0.45 μm Whatman cellulose
acetone filters (Song et al., 2013; Zhao et al., 2016a). The standards for dissolved total
carbon (DTC) were prepared from reagent grade potassium hydrogen phthalate in





ultra-pure water, while dissolved inorganic carbon (DIC) were determined using a
mixture of anhydrous sodium carbonate and sodium hydrogen carbonate. DOC was
calculated by subtracting DIC from DTC, both of which were measured using a Total
Organic Carbon Analyzer (TOC-VCPN, Shimadzu, Japan). Total nitrogen (TN) was
measured based on the absorption levels at 146 nm of water samples decomposed
with alkaline potassium peroxydisulfate. Total phosphorus (TP) was determined using
the molybdenum blue method after the samples were digested with potassium
peroxydisulfate (APHA, 1998). pH was measured using aPHS-3C pH meter at room
temperature ($20\pm2\,^{\circ}\mathrm{C}$).
**2.2 CDOM absorption measurement**
All water samples were filtered at low pressure at two steps: 1) filtered at low
pressure through a pre-combusted Whatman GF/F filter (0.7μm), and 2) further
filtered through pre-rinsed 25 mm Millipore membrane cellulose filter (0.22 μm).
Absorption spectra were obtained between 200 and 800 nm at 1 nm increment using a
Shimadzu UV-2600PC UV-Vis dual beam spectrophotometer (Shimadzu Inc., Japan)
through a 1 cm quartz cuvette (or 5 cm cuvette for ice melted water samples). Milli-Q
water was used as reference for CDOM absorption measurements. The Napierian
absorption coefficient ($a_{\mathrm{CDOM}}$) was calculated from the measured optical density ($OD$)
of samples using Eq. (1):

$$a_{CDOM}(\lambda) = 2.303[OD_{S(\lambda)} - OD_{(null)}]/\beta \tag{1}$$

where $\beta$ is the cuvette path length (0.01 or 0.05m) and 2.303 is the conversion factor
of base 10 to base $e$ logarithms. To remove the scattering effect from the limited fine





particles remained in the filtered solutions, a necessitated correction was implemented
by assuming the average optical density over 740–750 nm to be zero (Babin et al.,
2003). All absorption measurements were conducted within 48 h after the samples
were shipped back to the laboratory. In addition, $SUVA_{254}$ and $a_{CDOM}(250/365)$ were
calculated to characterize and group CDOM with respect to their compositional
features and try to link DOC based on CDOM grouping.

## 3. Results and discussion


### 3.1. Biological and geochemical characteristics


The biological and geochemical properties in the water bodies are diverse. Chl-a
concentrations (46.44±59.71 μg/L) changed from 0.28 to 521.12 μg/L, with the mean
of 46.44 μg/L. TN and TP concentration were very high in fresh lake water, saline
lake water and particularly urban water bodies (Table 1), indicating that most of the
waters are heavily eutrophic. It is worth noting that Chl-a concentration was still high
7.3±19.7 μg/L even in ice-covered lakes in winter from Northeast China, which
resulted from high TN (4.3±5.4 mg/L) and TP (0.7±0.6 mg/L) concentrations even
under ice cover. Electric conductivity (EC) and pH were high in the semi-arid and arid
regions, and they were 1067-41000 μs/cm and 7.1-11.4, respectively. This is due to
specific regional hydro-geologic and climatic conditions. The results are consistent
with previous findings (Song et al., 2013; Wen et al., 2016). Overall, waters were
highly turbid with high TSM concentrations (119.55 ±131.37 mg/L), but there were
big variation between different types of waters (Table 1). Hydrographic conditions
exerted strong impact on water turbidity and TSM concentration, thus these two




parameters of river and stream samples were excluded in this study (Table 1). Large
variations of water quality parameters in the extensive geographic area, for example
in China, provide a more comprehensive dataset for examining the relationship
between DOC and CDOM, and the result is very helpful for establishing remote
sensing models to estimate DOC through CDOM absorption properties (Cardille et al.,
2013; Zhu et al., 2014; Kutser et al., 2015).

**[Insert Table 1 about here]**

**3.2. DOC concentrations in different types of waters**
DOC concentrations changed remarkably in the investigated waters (Table 1). DOC
concentrations were low in rivers, while they were much lower in ice melting waters
sampled in winter, which is consistent with previous findings (Bezilie et al., 2002;
Shao et al., 2016). It should be noted that large variations were observed in water
samples from rivers and streams (Table 2) (Raymond and Saiers, 2010; Ward et al.,
2012), due to the strong connection with hydrological condition and catchment
landscape features (Neff et al., 2006; Agren et al., 2007; Lee et al., 2015). Generally,
low DOC concentrations were found in rivers or streams in the drainage systems in
Tibetan Plateau or arid regions in Northwest China where soil contains relative low
level of soil organic carbon, but the high DOC concentrations were found in rivers or
streams surrounded by forest or wetlands in Northeast China. The similar findings
were reported by Agren et al. (2007, 2010). Among the five types of waters, relatively
higher DOC concentrations, ranging from 2.3 to 300.6 mg/L, were found in many
saline lakes, in the Songnen Plain, the HulunBuir Plateau and some areas in Tibetan





Plateau (see Fig.2 for location), which is consistent with previous investigations
conducted in the semi-arid or arid regions (Curtis et al., 1995; Song et al., 2013; Wen
et al., 2016). However, some of saline lakes supplied by snow melt water or ground
water exhibited relatively lower DOC concentrations even with high salinity.
Compared with samples collected in growing seasons, higher DOC concentrations
(7.3-720 mg/L) were observed in ice-covered water bodies, due to the condensed
effect caused by the DOC discharged from ice formation (Bezilie et al., 2002; Shao et
al., 2016). This condensed effect was particularly marked in these shallow water
bodies where ice forming remarkably condensed the DOC in the underlying waters
(Zhao et al., 2016a). Even in rivers or saline lakes, the concentrations of DOC
demonstrated obvious variations (Table 2). Comparatively, rivers from Qinghai
exhibited lower DOC concentration, while these from the Liaohe and Inner Mongolia
showed much higher DOC concentration (Table 2). Similarly, large DOC variations
were observed in saline lakes in different regions (Table 2). Much higher DOC
concentrations were found in saline lakes in Qinghai and Hulunbir, while relative low
concentrations were observed in Xilinguole Plateau and the Songnen Plain.

**[Insert Table 2 about here]**

**3.3. DOC versus CDOM for various types of waters**
*3.3.1 Freshwater lakes and reservoirs*
The relationship between DOC and CDOM has been researched based on CDOM
absorption spectra at different wavelengths (Fichot and Benner, 2011; Spencer et al.,
2012; Song et al., 2013; Brezonik et al., 2015). As suggested by Fichot and Benner





(2011), CDOM absorptions at 275 nm ($a_{CDOM}275$) and 295 nm ($a_{CDOM}295$) have
stable performances for DOC estimates for coastal waters. In current study, a strong
relationship ($R^2 = 0.85$) between DOC and $a_{CDOM}(275)$ was found in fresh lakes and
reservoirs (Fig.3a). However, the participation of $a_{CDOM}(295)$ explains very limited
variance, thus it is not considered in the regression models. Regression analyses of
water samples collected from different regions indicated that the slopes varied from
1.30 to 3.13 (Table 3). Water samples collected from East China and South China had
lower regression slope values (Table 3), and lakes and reservoirs were generally
mesotrophic or eutrophic (Huang et al., 2014; Yang et al. 2012, and references
therein). Phytoplankton degradation may contribute relative large portion of CDOM
and DOC in these water bodies (Zhang et al., 2010), due to the lower molecular
weight, its absorption is different from that derived from terrestrial systems (Helms et
al., 2008). Comparatively, fresh waters in Northeast and North China revealed larger
regression slopes (Table 3).Waters in Northeast China are surrounded by forest,
wetlands and grassland and therefore they generally exhibited high proportion of
colored fractions, CDOM (Helms et al., 2008). Soils in Northeast China are rich in
organic carbon, which may also contribute to high concentration of DOC and CDOM
in waters in this region (Jin et al., 2016; Zhao et al., 2016a). Compared with waters in
East and South China, waters in Northeast China showed less algal bloom due to low
temperature, thus autochthonous CDOM was less presented in waters in Northeast
China (Song et al., 2013; Zhao et al., 2016a). As suggested by Brezonik et al. (2015)
and Cardille et al. (2013), CDOM in the eutrophic waters or those with very short



resident time may show seasonal variation due to algal bloom or hydrological
variability, while CDOM in some oligotrhopic lakes or those with long resident time
may show an opposite pattern.

**[Insert Table 3 about here]**

**[Insert Fig.3 about here]**

*3.3.2 Saline lakes*
A strong relationship between DOC and $a_{CDOM}(275)$ ($R^2 = 0.85$) was demonstrated for
saline lakes (Fig.3b). However, compared to fresh waters, much lower regression
slope value (slope = 1.28) was found in saline lakes. Similar to fresh waters, the
slopes of most saline lakes exhibited large variations between different regions (Table
3), ranging from 0.86 in Tibetan waters to 2.83 in Songnen Plain waters (see Fig.2 for
location). As the extreme case, the slope value was only 0.33 as demonstrated in the
embedded diagram in Fig.3b. Saline lakes in semi-arid or arid regions generally
exhibit higher regression slope values, for example, west Songnen Plain (2.83),
Hulunbir Plateau and East Inner Mongolia Plateau (1.79). Whereas, waters in the west
Inner Mongolia Plateau (1.13), the Tibetan Plateau (0.86) exhibited low slope values
(Table 3), and the extreme low value was measured in the Lake Qinhai in Tibetan
Plateau. Lakes in Tarim Basin were affected by strong photo-bleaching, due to the
long resident time and strong solar radiation (Spencer et al., 2012; Song et al., 2013;
Wen et al., 2016). Thereby, smaller regression slopes were found and less colored
portion of DOC was presented in waters in semi-arid to arid regions, especially for
these closed lakes with enhanced photochemical processes (Spencer et al., 2012; Song




et al., 2013; Wen et al., 2016). The findings highlighted the difference in remote
sensing of DOC through CDOM absorption algorithm between saline and fresh lakes,
thereby different models should be established to accurately estimate DOC in waters
(Cardille et al., 2013; Brezonik et al., 2015).
### *3.3.3 Streams and rivers*
Although some of the samples scattered from the regression line (Fig.3c), close
relationship between DOC and $a_{CDOM}$(275) was found for samples collected in rivers
and streams. Compared with the other water types (Fig.3), rivers and streams
exhibited the highest regression slope value (slope = 3.13). Further regression analysis
with water samples sub-datasets collected in different regions indicated that slope
values presented large variability, ranging from 1.07 to 8.49. The lower regression
slope values were recorded in water samples collected in rivers and stream in
semi-arid and arid regions, such as the Tibetan Plateau, Mongolia Plateau and Tarim
Basin, while the higher values were found in samples collected in streams originated
from wetland and forest in Northeast China (Table 3). Rivers and streams in North,
East and South China generally exhibited intermediate values. In addition, water
samples in large river generally presented relatively low slope value; streams,
especially head water originating from forest and wetland dominated regions show
higher regression slope values (e.g., Branches from the Nenjiang and the Songhua
River in Table 3), which is consistent with the findings from Helm et al. (2008) and
Spencer et al. (2012). In fact, landscape pattern and soil organic carbon in the
catchment are important factors governing the terrestrial DOC and CDOM




characteristics in rivers and streams (Wilson and Xenopoulos, 2008; Jaffe et al., 2008;
Agren et al., 2010; Lai et al., 2016).

DOC concentration is strongly associated with hydrological conditions (Neff et al.

2006; Agren et al. 2007). Thereby, the relationships between CDOM and DOC in
river and stream waters are very variable (Lee et al., 2015) due to the hydrological
variability and catchment features (Agren et al., 2010; Spencer et al., 2009; 2012). To
investigate the dynamics of CDOM absorption and DOC concentrations, three
sections were investigated in three major rivers in Northeast China (see Figure S1 for
location). River flow exerted obvious effect on DOC and CDOM (Fig.4) and flood
impulse brought large amount of DOC and CDOM into river channels, which is
consistent with previous findings (Neff et al., 2006; Larson et al., 2007). As shown in
Fig.4, the relationship between river flows and DOC is rather complicated, which is
mainly caused by the land use, soil properties, relief, slope, the proportion of wetlands
and forest, climate and hydrology of the catchments (Neff et al., 2006; Sobek et al.,
2007; Spencer et al., 2012; Zhou et al., 2016), with additional influence by sewage
discharge into rivers. The relationships between DOC and $a_{CDOM}(275)$ in sections
along three rivers in Northeast China were demonstrated in Fig.5. The sampling point
in the Yalu River is near the river head source, thus strong relationship was exhibited
with large slope (Fig.5a), due to that the DOC and CDOM were fresh and less
disturbed by pollution from anthropogenic activities (Spencer et al., 2012; Shao et al.,
2016). The relationship between DOC and $a_{CDOM}(275)$ in the Songhua River at Harbin
City section was much scattered (Fig.5c) and this is mainly attributed to both point





and non-point source pollution that cause the composition and colored fractions of
DOC and DOM much varied comparing to river head waters with less human
disturbance. Similar mechanisms are further detailed in section 3.3.4 with urban
waters. With respect to Fig.5b, it is an in-between case. The sampling point was
affected by effluent from Baishan City, thus the coefficient of determination ($R^2$=
0.822) and the regression slope (3.72) were lower than that from the Yalu River at
Changbai point, while higher than that from the Songhua River at Harbin point.
Thereby, both spatial and temporal changes of the relationships between DOC and
CDOM were observed, and anthropogenic activities further complicated the
relationship.

**[Insert Fig.4 and Fig.5 about here]**

*3.3.4 Urban waters*
Relative close relationship between DOC and $a_{CDOM}(275)$ was revealed in urban
waters (Fig.3d, $R^2$= 0.71), where it was much scattered compared with other water
types (Fig.3), particularly for water samples with DOC concentration less than 60
mg/L. Similarly, regression slope values changed remarkably, ranging from 0.87 to
2.45. It is apparent that urban waters are severely impacted by human activities,
particularly sewage, effluents and runoff from urban impervious surface containing
large amount of DOM (Yang et al. 2008; Zhao et al., 2016b, and references therein).
High nutrients also usually result in algal bloom in most urban water bodies (Chl-a
range: 1.0-521.1 μg/L; average: 38.9 μg/L). Thereby, DOC and CDOM derived from
phytoplankton may also contribute a portion that should not be neglected (Xing et al.





2006; Zhang et al., 2010; Zhao et al., 2016b). More or less affected by sewage
effluent, the DOM in urban waters is much complex than those from natural water
bodies. Thus, a large variation of the relationship between DOC and $a_{CDOM}(275)$ was
found in urban waters.
*3.3.5 Ice covered lakes and reservoirs*
The closest relationship ($R^2 = 0.93$) between DOC and $a_{CDOM}(275)$ was recorded in
waters beneath ice covered lakes and reservoirs in Northeast China (Fig.3e). It was
argued that the close relationship indicated the concurrent processes taken place for
DOC accumulation and CDOM biogeochemical activities (Finlay et al., 2003;
Stedmon et al., 2011). The strong positive correlations between DOC and $a_{CDOM}275$ is
probably due to ice formation condensed these two parameters. The other possible
explanation was that ice and snow cover shielded out most of the solar radiation that
might cause a series of biochemical process for CDOM contained in water; further,
the inflows and direct rainfall over lakes or reservoirs also diminished, thus causing
limited effect on DOC and CDOM composition (Uusikiv et al., 2010; Belzile et al.,
2002). Further, the autochthonous DOC and CDOM in ice covered waters were also
limited due to the relatively weak primary production in winter (Chl-a = 7.3 μg/L),
resulting in much close relationship in winter waters.
Comparatively, a weak relationship between DOC and $a_{CDOM}(275)$ was
demonstrated in ice melting waters (Fig.3f), which was probably due to the ice/water
depth ratio causing variation of dissolved components expelled during ice formation.
The other reason is the biologically derived DOC in the ice matrix, which changes





with the variation of light and nutrient (Arrigo et al., 2010; Zhang et al., 2010).
Apparently, CDOM from ice melting waters were mainly originated from maternal
water during the ice formation, also from algal biological processes (Stedmon et al.,
2011; Arrigo et al., 2010). Similarly, snow cover, and nutrients in the ice also causes
the variation of biochemical processes that ultimately complicate the relationship
between DOC and CDOM (Bezilie et al., 2002; Spencer et al., 2009). Interestingly,
the regression slopes for ice samples (1.35) and under lying water sample (1.27) are
very close. In addition, there was a significant relationship between DOC in ice and
underlying waters ($R^2 = 0.86$), indicating the dominant components of CDOM and
DOC in the ice are from maternal underlying waters.
**3.3.6 DOC versus $a_{CDOM}(440)$**
CDOM absorption at 440 nm, i.e., $a_{CDOM}(440)$, is usually used as a surrogate to
represent its concentration (Bricaud et al., 1981; Babin et al., 2003), and widely used
in remote sensing community to quantify CDOM in waters (Lee et al., 2002; Binding
et al., 2008; Zhu et al., 2014). Significant relationships between DOC and $a_{CDOM}(440)$
were found in different types of waters (Fig.5). Compared to DOC versus $a_{CDOM}(275)$,
the relationships were more scattered due to the weak CDOM absorption at longer
wavelength (Bricaud et al., 1981; Binding et al., 2008). Through comparing Fig.3with
Fig.6, it can be found that the overall relationships between DOC and CDOM at 440
nm resemble that at 275 nm for different types of waters. This has important
implication for remote sensing of DOC through the CDOM absorption as a bridge
(Zhu et al., 2014; Kuster et al., 2015; Brezonik et al., 2015). It is also worth noting



that most of the streams and rivers, including some of the urban water bodies, are not
suitable to quantify DOC through remote sensing imagery, and this is due to that
medium or even coarse resolution imagery cannot effectively capture the change of
signals from these small water bodies. However, the systematic examination for the
relationship between DOC and CDOM may help to quantify DOC through CDOM
absorption for deploying portable sensors in streams or rivers that can measure
CDOM absorption more accurately with dynamic manner (Spencer et al., 2012; Lee et
al., 2015; Ruhala and Zarnetske, 2016).

**[Insert Fig.6 about here]**

**3.4 CDOM molecular weight and aromacity versus DOC**
**3.4.1 CDOM versus SUVA254 and $a_{CDOM}(250/365)$**
The large variations of the slope of regression of DOC and $a_{CDOM}(275)$ in different
types of waters are probably due to the aromacity and colored fractions in DOC
component (Spencer et al., 2009, 2012; Lee et al., 2015). $SUVA_{254}$ is an effective
indicator to characterize CDOM molecular weight. It may reflect the regression slope
value between DOC and CDOM absorption at 275 nm. It is obvious that $SUVA_{254}$ had
high values in fresh lakes, and waters from rivers or streams as well (Fig.7a). Saline
water and ice covered waters in Northeast China showed intermediate $SUVA_{254}$
values, while urban water and ice melting water exhibited lower values. The M value,
i.e., $a_{CDOM}(250/365)$ is another indicator to demonstrate the variation of molecular
weight and aromacity of CDOM components (De Haan, 1993). Fresh lake water, river





and stream water, and urban water exhibited low M values (Fig.7b), which indicated
that larger aromacity dominant for these three types of waters. Saline water, ice
covered water in Northeast China and ice melting water showed higher M values.
Since SUVA$_{254}$ is a proxy based on the ratio to DOC, it is inappropriate to establish
the relationship between CDOM and DOC based on the SUVA$_{254}$ classification.
Thereby, only M values, which reveal molecular weight and aromacity, might help to
estimate DOC through CDOM absorption based on M threshold values for various
types of waters.

**[Insert Fig.7 about here]**

*3.4.2 Regression based on M values*
Regression models between DOC and a$_{CDOM}$(275) were established based on M
threshold values, which were determined through trial test with respect to the
concentrations of DOC versus a$_{CDOM}$(275). A relative weaker relationship between
DOC and a$_{CDOM}$(275) was revealed in dataset where M values were less than 5
(Fig.8a). It should be noted that the high regression slope values appeared indifferent
groups of subset data (Fig.8a-h). The large range of M value (0<M<4.0) may explain
the scattered data pairs in Fig.8a and this is also the reason for the group with M
values ranging from 4 to 6 (Fig.8b). Better regression models appeared in groups with
intermediate M values (Fig.8c-f), with small range of regression slope values (1.15 -
1.38) and high determination of coefficients (R$^2$> 0.88). Regression slope values
decreased with the increasing of M values (Fig.8g-h). Weak relationship between
DOC and a$_{CDOM}$275 appeared with relative lower or higher M values (Fig.8g). Very





significant relationship ($R^2 = 0.99$) was found with extremely high M values (Fig.8h).
Most of samples collected from these groups were presented in the embedded diagram
in Fig.3b, and the limited water bodies in the group may explain this coincidently high
$R^2$ value. With more samples collected from different water bodies in this extreme
group, a weak relationship between DOC and $a_{CDOM}(275)$ may appear, while future
explorations are needed.
As noted in Fig.8c-f, close regression slopes implicate that a comprehensive
regression model with intermediate M value groups may be achieved. As expected, a
promising regression model (the diagram was not shown) between DOC and
$a_{CDOM}275$ was achieved (y = 1.269x + 6.55, $R^2 = 0.925$, N = 998, p < 0.001) with
pooled dataset shown in Figs.8c to 8f. Inspired by this idea, the relationship between
CDOM and DOC also examined with pooled data. As shown in Fig.9a, a significant
relationship between DOC and $a_{CDOM}(275)$ was obtained with the pooled dataset (N =
1504) collected from different types of inland waters. However, it should be noted
that the extremely high DOC samples may advantageously contribute the better
performance of the regression model. Thus, regression model excluding these eight
samples (DOC > 300 mg/L) was still significant (Fig.9b, $R^2 = 0.66$, p < 0.01). In
addition, regression model based on logarithm transformed data was established
(Fig.9c, $R^2 = 0.82$, p < 0.01). Most of the paired data sitting close to the regression
line except some scattered ones. This also implies that relative accurate regression
model for CDOM versus DOC can be achieved with data collected in inland waters at
global scale (Sobek et al., 2007), which might be helpful in quantifying DOC through





linking with CDOM absorption spectra, and the latter parameter can be estimated
from remote sensing data (Zhu et al., 2011; Kuster et al., 2015).

**[Insert Fig.8 and Fig.9 about here]**

## 4. Conclusions

As a powerful technology, remote sensing plays a crucial role in assessing CDOM and
DOC in water environment. In order to get accurate estimates of CDOM and DOC in
waters, it is necessary to get insight into the regional water optical properties for
developing semi-analytical or analytical models with remotely sensed data. Based on
the measurement of CDOM absorption spectral and DOC laboratory analysis, we
have systematically examined the relationships between CDOM and DOC in various
types of waters in China. This investigation showed that CDOM absorption varied
significantly. River waters and fresh lake waters exhibited high CDOM absorption
values and specific CDOM absorption ($SUVA_{254}$). On the contrast, saline lakes
illustrated low $SUVA_{254}$ values due to the long residence time and strong
photo-bleaching effects on waters in the semi-arid regions. Influenced by effluents
and sewage waters, CDOM from urban water bodies showed much complex
absorption feature. $SUVA_{254}$ for CDOM was lowest in ice melting water samples.

The current investigation indicated that the relationships between CDOM

absorption and DOC varied remarkably by showing very varied slope values of
regression models in various types of waters. The slope values of saline lakes and
urban waters were close to unity, slope values of river water were highest (~ 3.1), and
slope values of other water types were in between. It should also be highlighted that





head river water generally exhibit larger regression slope values, while rivers affected
by anthropogenic activities show lower slope values. When all the data set were
pooled together, the slope for regression model was about 1.3, but with much bigger
uncertainty ($R^2$ = 0.66). The accuracy of regression model between $a_{CDOM}(275)$ and
DOC was improved when CDOM absorptions were divided into different sub-groups
according to M values. Our finding highlights that remote sensing models for DOC
estimation based on the relationship between CDOM and DOC should consider water
types or cluster waters into several groups according to their absorption features.
More researches are still needed to further improved model accuracy.

**Acknowledgements**
The authors would like to thank financial supports from the National Basic Research
Program of China (No. 2013CB430401), Natural Science Foundation of China
(No.41471290), and "One Hundred Talents" Program from Chinese Academy of
Sciences granted to Dr. Kaishan Song. Thanks are also extended to all the staff and
students for their efforts in field data collection and laboratory analysis, and Dr. Hong
Yang to review and polish the English language. Last but not the least, the authors
would like to thank the editor and two anonymous referees for their valuable
comments that really help a lot in improving the manuscript.

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




**Figures**
Fig.1. the diagram shows the regulating factors that influence the relationship between
CDOM and DOC.

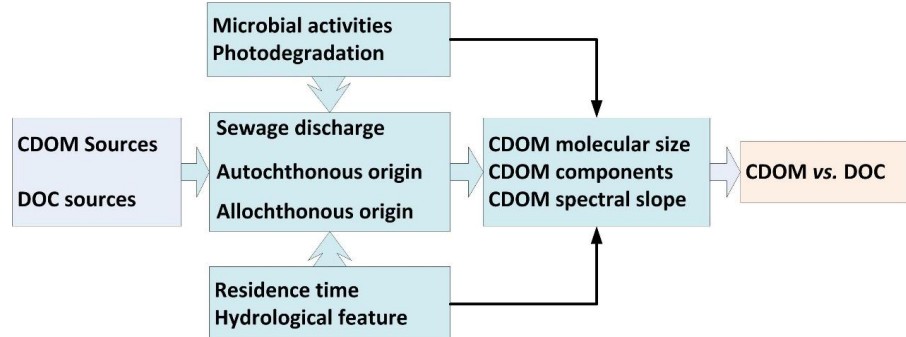























Fig.2. Water types and sample distributions across the mainland China. The dash line
shows the boundary of some typical geographic units.

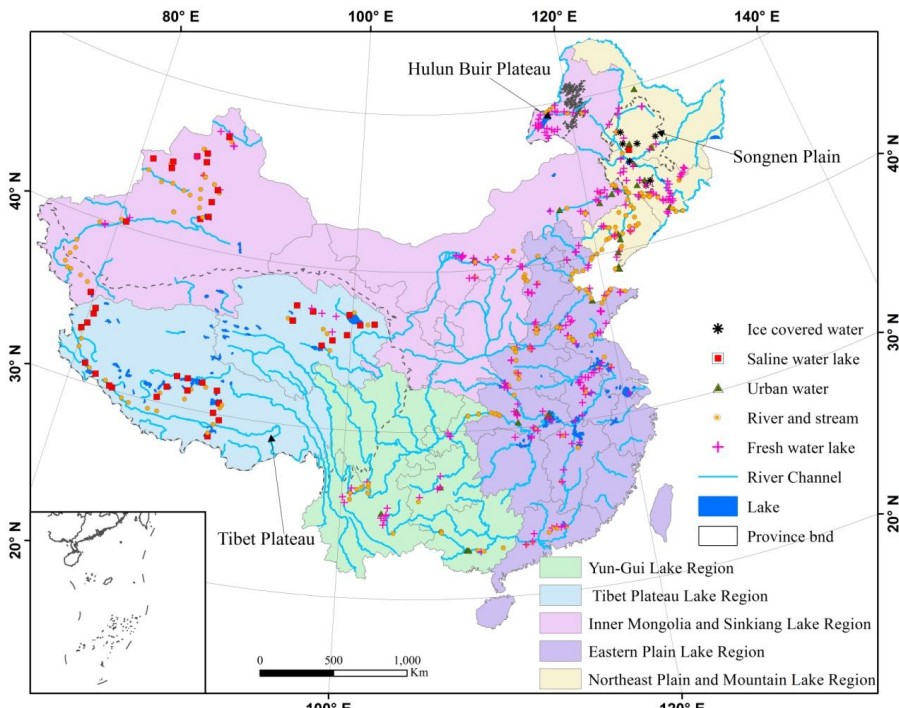




Fig.3. Relationship between DOC and $a_{CDOM}(275)$ in different types of inland waters,
(a) fresh water lakes, (b) saline water lakes, (c) river and stream waters, (d) urban
waters, (e) ice covered lake underlying waters, and (f) ice melting lake waters.








Fig.4. Flow dynamics for three rivers in Northeast China and corresponding DOC and
CDOM variations; (a) the Yalu River near Changbai County, (b) the Hunjiang River
with DOC and CDOM sampled at Baishan City, while the river flow gauge station is
near the Tonghua City, (c) the Songhua River at Harbin City. Note, the flow data for
the Yalu River and the Hunjiang River were the average values measured during
1970s, while the Songhua River was measured during 2000-2010.

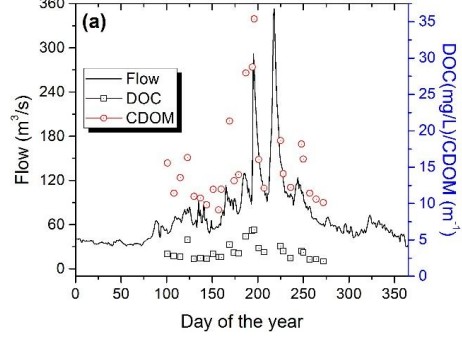


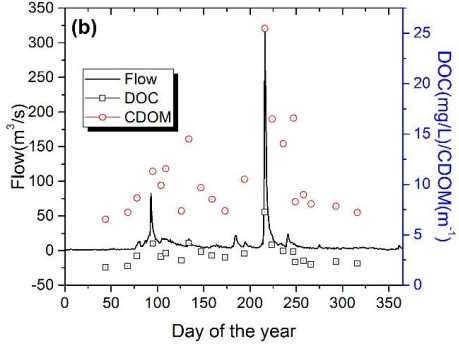


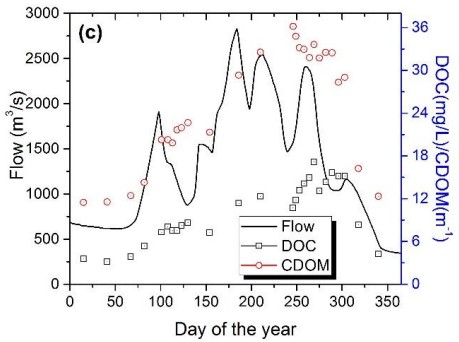






Fig.5. The relationships between $a_{CDOM}275$ and DOC at sections across (a) the Yalu
River, (b) the Hunjiang River, and (c) the Songhua River. The samples were collected
at each station at about one week or around ten days in ice free season in 2015.

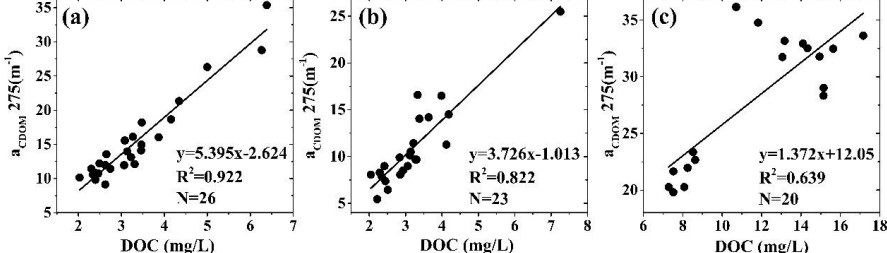

























Fig.6. Relationship between DOC and $a_{CDOM}$(440) in different types of inland waters,
(a) fresh water lakes, (b) saline water lakes, (c) river and stream waters, (d) urban
waters, (e) ice covered lake underlying waters, and (f) ice melting waters.








Fig.7. Comparison of (a) SUVA254, and (b) M ($a_{250}:a_{365}$) values in various types of
inland waters. FW, fresh lake water; SW, saline lake water, RW, river or stream water;
UW, urban water; WW, ice covered winter water from Northeast China; IMW, ice
melt water from Northeast China.

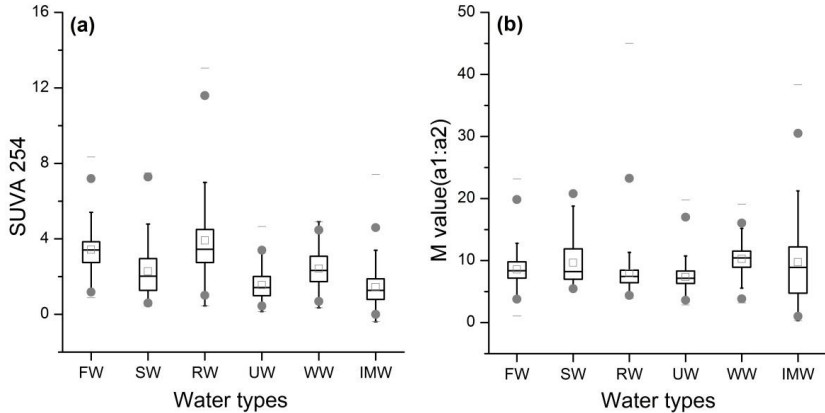





Fig.8. Relationship between DOC and $a_{CDOM}275$ sorted by M ($a_{CDOM}250/365$) values
ranges, (a) M <4.0, (b) 4.0< M<6.0, (c) 6.0< M< 7.0, (d) 7.0< M< 8.0, (e) 8.0< M<
10.0, (f) 10.0< M< 12.0, (g) 12.0< M< 20.0, and (h) 20.0< M< 68.0.

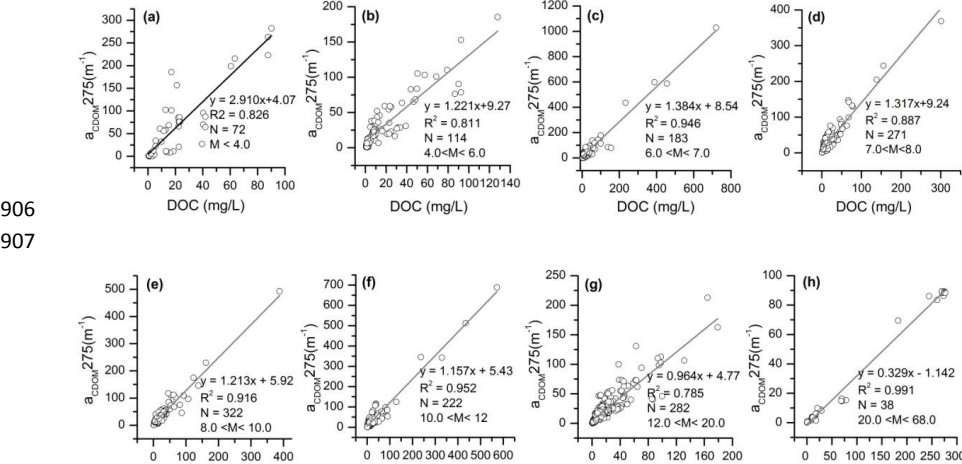



Fig.9. the relationships between $a_{CDOM}275$ and DOC concentrations. (a) regression
model with pooled dataset; (b) regression model with DOC concentration less than
300 mg/L; (c) regression model with natural logarithmic transformed data.

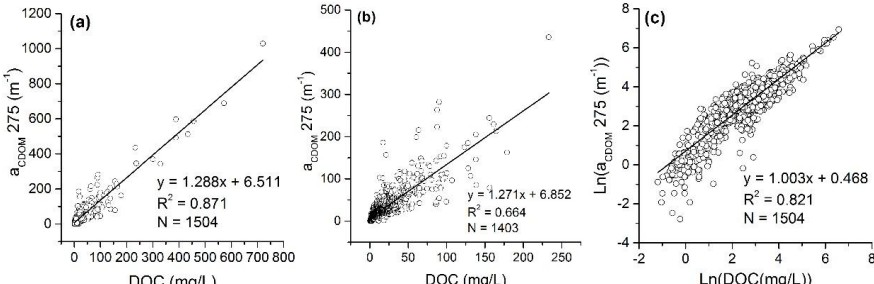






















**Tables**

Table 1. Water quality in different types of waters, DOC, dissolved organic carbon;
EC, electrical conductivity; TP, total phosphorus; TN, total nitrogen; TSM, total
suspended matter; Chl-a, chlorophyll-a concentration.

|  |  | DOC (mg/L) | EC μs/cm | pH | TP (mg/L) | TN (mg/L) | TSM (mg/L) | Chl-a (μg/L) |
|---|---|---|---|---|---|---|---|---|
| FW | Mean | 10.2 | 434.0 | 8.2 | 0.5 | 1.6 | 67.8 | 78.5 |
|  | Range | 1.9-90.2 | 72.7-1181.5 | 6.9-9.3 | 0.01-10.4 | 0.001-9.5 | 0-1615 | 1.4-338.5 |
| SW | Mean | 27.3 | 4109.4 | 8.6 | 0.4 | 1.4 | 115.7 | 9.0 |
|  | Range | 2.3-300.6 | 1067-41000 | 7.1-11.4 | 0.01-6.3 | 0.6-11.0 | 1.4-2188 | 0-113.7 |
| RW | Mean | 8.3 | 10489.1 | 7.8-9.5 | - | - | - | - |
|  | Range | 0.9-90.2 | 3.7-1000 | 8.6 | - | - | - | - |
| UW | Mean | 19.44 | 525.4 | 8.0 | 3.4 | 3.5 | 50.5 | 38.9 |
|  | Range | 3.5-123.3 | 28.6-1525 | 6.4-9.2 | 0.03-32.4 | 0.04-41.9 | 1-688 | 1.0-521.1 |
| WW | Mean | 67.0 | 1387.6 | 8.1 | 0.7 | 4.3 | 181.5 | 7.3 |
|  | Range | 7.3-720 | 139-15080 | 7.0-9.7 | 0.1-4.8 | 0.5-48 | 9.0-2174 | 1.0-159.4 |
| IMW | Mean | 6.7 | 242.8 | 8.3 | 0.19 | 1.1 | 17.4 | 1.1 |
|  | Range | 0.3-76.5 | 1.5-4350 | 6.7-10 | 0.02-2.9 | 0.3-8.6 | 0.3-254.6 | 0.28-5.8 |


Note: FW, fresh water lake; SW, saline water lake, RW, river or stream water; UW, urban water;
WW, ice covered winter water from Northeast China; IMW, ice melt water from Northeast China.



Table 2. Descriptive statistics of dissolved organic carbon (DOC) and $a_{\text{CDOM}}(440)$ in
various types of waters.

| Type | Region | DOC | | | | $a_{\text{CDOM}}(440)$ | | | |
|---|---|---|---|---|---|---|---|---|---|
| | | Min | Max | Mean | S.D | Min | Max | Mean | S.D |
| River | Liaohe | 3.6 | 48.2 | 14.3 | 9.49 | 0.46 | 3.68 | 0.92 | 0.58 |
| | Qinghai | 1.2 | 8.5 | 4.4 | 1.96 | 0.13 | 2.11 | 0.54 | 0.63 |
| | Inner M | 16.9 | 90.2 | 40.4 | 24.84 | 0.32 | 7.46 | 1.03 | 2.11 |
| | Songhua | 0.9 | 21.1 | 8.1 | 4.96 | 0.32 | 18.93 | 3.2 | 4.19 |
| | | | | | | | | | |
| Saline | Qinghai | 1.7 | 130.9 | 67.9 | 56.7 | 0.13 | 0.86 | 0.36 | 0.23 |
| | Hulunbir | 8.4 | 300.6 | 68.5 | 69.2 | 0.82 | 26.21 | 4.41 | 4.45 |
| | Xilinguo | 3.74 | 45.4 | 14.2 | 8.8 | 0.36 | 4.7 | 1.34 | 0.88 |
| | Songnen | 3.6 | 32.6 | 16.4 | 7.4 | 0.46 | 33.80 | 2.4 | 3.78 |






Table 3. Fitting equations for DOC against $a_{CDOM}(275)$ in different types of waters
except ice covered lake underlying water and ice melting waters.

| Water types | Region or Basin | Equations | $R^2$ | N |
|---|---|---|---|---|
| Freshwater lakes | Northeast Lake Zone | y = 3.13x-3.438 | 0.87 | 102 |
| | North Lake Zone | y = 2.16x-1.279 | 0.90 | 63 |
| | East Lake Zone | y = 1.98x+7.813 | 0.66 | 69 |
| | Yungui Lake Zone | y = 1.295x-44.56 | 0.71 | 54 |
| Saline lakes | Songnen Plain | y = 2.383x+1.101 | 0.92 | 159 |
| | East Mongolia | y = 1.791x+8.560 | 0.67 | 57 |
| | West Mongolia | y = 1.133x+3.900 | 0.81 | 46 |
| | Tibetan Plateau | y = 0.864x+2.255 | 0.84 | 83 |
| Rivers or streams | Branch of the Nenjiang River | y = 7.655x-42.64 | 0.81 | 33 |
| | Songhua River stem | y = 3.759x-6.618 | 0.71 | 29 |
| | Branch of Songhua River | y = 8.496x-12.14 | 0.98 | 33 |
| | Liao River Autumn 2012 | y = 1.099x+3.900 | 0.80 | 38 |
| | Liao River Autumn 2013 | y = 1.073x-4.157 | 0.88 | 28 |
| | Liao River Spring 2013 | y = 2.262x-10.32 | 0.85 | 25 |
| | Rivers from North China | y = 3.154x-1.207 | 0.87 | 48 |
| | Rivers from East China | y = 3.037x-2.585 | 0.88 | 47 |
| | Rivers from Tibetan | y = 2.345x+2.375 | 0.87 | 41 |
| Urban waters | Waters from Changchun | y = 2.471x-2.231 | 0.54 | 48 |
| | Waters from Harbin | y = 1.413x-4.521 | 0.67 | 31 |
| | Waters from Beijing | y = 0.874x+11.12 | 0.63 | 27 |
| | Waters from Tianjin | y = 0.994x + 7.368 | 0.57 | 23 |


