# Peer review of "A systematic examination of the relationships between CDOM and"

_Hydrology and Earth System Sciences, 2017_

## Referee Comment (RC1) · Anonymous Referee #1 · 1 May 2017

This paper presents a series of regression equations between DOC concentrations and optical properties of the DOM across a range of water bodies in China. The authors have amassed an impressive data set, and applying this data set to questions of DOC biogeochemistry could make a useful contribution. Unfortunately the paper, as currently written, has some flaws that limit its value.

The paper focuses on two objectives stated in the Introduction, and a third objective that, for some reason, is presented in the Methods (lines 157-159). The objectives all are targeted at examining the relationship between DOC concentrations and optical properties, particularly absorbance at 275 nm or 440 nm. The paper would be improved if it were structured around testable hypotheses, which I think the authors could do

without too much additional work.

The primary means of data analysis is simple linear regression, and it appears that perhaps multiple linear regression was attempted (line 279-280). Surprisingly, no description of data analysis is provided in the paper (or the supplemental information). In fact, P values are not even provided for the regression analyses. Nor is there any indication of testing for normality or other assumptions for linear regression. Many of the graphs show that a single data point, or a couple data points, appears to be leveraging the relationship (e.g., Fig 3c, Fig 3e, Fig 3f, Fig 6d, Fig 6f, and others). In these cases, the validity of the regression equation is highly questionable.

In the case of Fig 8, it is not clear how the groupings were selected. The text mentions "trial and error" which suggests to be it was a very subjective process of selecting the M ranges for the groups.

I am a bit concerned about the holding time (up to 2 days) before filtration. Do the authors have any evidence that there was no degradation of DOC during the holding time? Some concern for chlorophyll-a. Also, it is questionable to collect and store DOC for optical analysis in HDPE bottles. Why was HDPE used instead of glass?

In the end, the authors state that SUVA is not an appropriate metric for the purposes of their study because its calculation includes DOC concentration. This left me wondering why it was included at all?

I think the Introduction could be shortened by as much as a third without any loss. Much of the introduction deals with remote sensing for DOC, but this paper does not address remote sensing directly; the background information on remote sensing could be greatly reduced within the Introduction and also the Discussion. I think developing some testable hypotheses and keeping the Introduction (and the whole paper) focused narrowly on those hypotheses would make for a shorter, and more readable, paper.

I would strongly suggest separate Results and Discussion sections. As I read the paper

it was not always clear when the authors were making statements based on their data, versus general statements from literature.

Try to avoid vague statements such as "massive organic matter" (line 22) and "big variation" (line 230). The English in the paper is mostly correct, but it could certainly be improved if edited closely by a native English speaker.
* * *

---

## Short Comment (SC1) · 2 May 2017

Very impressive dataset and a very valuable research from remote sensing prospective. I definitely recommend to publish the paper after some minor modifications have been made.

The Authors mention in the text that using remote sensing for small ponds and lakes is problematic because of lack of appropriate remote sensing sensors. This may have been true some time ago (Palmer et al. 2015), but is not any more. Sentinel-2 imagery with 10 m spatial resolution is available globally. This kind of resolution is suitable for almost any pond, not speaking about lakes. Sentinel-2A data has been already used in mapping lake CDOM and DOC (Toming et al. 2016). Sentinel-2B was launched

two months ago and is currently in testing phase. Meaning that in a few months 10 m spatial resolution imagery will be available with 5 days revisit time at the equator and about 2-3 day revisit time for most lakes in China. Besides that Landsat-8 imagery with 30 m spatial resolution is also available. There have been several papers recently showing the usefulness of Landsat-8 in mapping lake CDOM/DOC. Consequently, the image data is not a problem anymore. This strengthens the value of this research even more. I recommend to improve the remote sensing part of the manuscript showing that there is plenty of data available now free of charge with very high spatial and temporal resolution and your study will help to improve usefulness of this data at very local to global scales.

SUVA is an important parameter used to describe carbon quality (e.g. in drinking water industry). Therefore, it is important to link remote sensing and SUVA more closely in the manuscript. Remote sensing of SUVA has been demonstrated at least in one recent paper cited several times by the Authors. I would recommend to add this reference in the 3.4 and strengthen the link between SUVA and remote sensing there.

Is there any information available for seasonal variability? At least in boreal zone CDOM decreases from spring to summer and then starts to increase again (e.g. Kutser 2012), but how about the CDOM-DOC or DOC-SUVA relationships? This would be a very interesting piece of information.

In general the paper is written well. There are some minor errors in names (e.g. must be Gulf of Finland not Finish Gulf in row 119) and some sentences can be modified, but the text is easily readable.

---

## Editor Comment (EC1) · C. Stamm (Editor) · 2 May 2017

Dear T. Kutser

thank you for participating in the discussion of this paper. In your comment you refer to a number of papers but you did not provide the references. Can you add this information? This would be useful for other readers.

Thanks a lot

Christian Stamm

Editor HESS

---

## Editor Comment (EC2) · C. Stamm (Editor) · 15 May 2017

Editor comment

HESSD-Manuscript **"A systematic examination of the relationships between CDOM and DOC in inland waters in China"** (HESS 2017-179).

Dear Dr. Y. Song

Reading through the manuscript, I came across a number of aspects on which I'd like

to comment on during this discussion phase.

**Comments on content:**

**L. 70 - 75:** Provide explanations on mechanisms how hydrology affects DOC and CDOM properties. Why shall catchment size *per se* be important? Can you explain why size is influential apart from affecting for example travel times?

**L. 316:** If you know about these influencing factors, why you cannot derive an explanatory model for the slope?

**L. 345 - 347:** Again, you should explain how hydrology and catchment characteristics can influence CDOM and DOC.

**L. 350 - 357, Fig. 4:** This part is highly misleading because the text evokes the impression that you compare DOC and CDOM to simultaneous measurements of discharge. Unfortunately, this is not the case. You write in the manuscript that the hydrographs correspond to long term averages and do not represent the actual flow conditions during the periods of your sampling campaign. However, you do not pay attention to that basic fact when displaying the data: By plotting discharge and concentrations against the same time axis (Fig. 4) you give the impression that a flow rate value on day X is linked to the concentrations for the same day. However, this is not true. Therefore, this way of presenting and interpreting the data is not acceptable. Such figures would only be correct if you can provide the discharge data from the years of sampling.

Should it not be possible to get access to this data, you have to adapt you data analysis accordingly. Because you neither have information about the actual flow for a given sampling date nor about the actual sequence of actual discharge from day to day you should not plot the data against the time axis. Instead you plot for

example the concentrations against a selected quantile of flow rates for the corresponding Julian day. This would also be more to the point because you argue that there is a relationship between flow rate and DOC/CDOM concentrations irrespective of the Julian day.

**L. 425 - 426, Fig. 3, Fig. 6:** The data sets in the two figures seem not be fully consistent. When comparing for example Fig. 3d and 6d, there are 3 – 4 data points with DOC concentrations of 80 – 120 mg $L^{-1}$ in Fig. 3d that are absent in Fig. 6d. How does it come? The same holds in the opposite direction with data of about 45 mg $L^{-1}$ in Fig. 6c. What is the explanation?

**L. 451:** I cannot see this low M values in Fig. 7B. Can you support your statement by a statistical metric?

**L. 462:** Your selection of categories for M is rather arbitrary. Additionally, when looking at them in their entirety, it is obvious that there is a general pattern in that the slope decreases with increasing M (see figure in the attachment). Hence, the slope simultaneously depends on DOC and M. Because M is simply the ratio between $a_{CDOM}250$ and $a_{CDOM}365$, it follows that $a_{CDOM}275$ is a function of DOC, $a_{CDOM}250$, and $a_{CDOM}365$. Instead of introducing first M and then classify M into categories, you better directly express $a_{CDOM}275$ as a function of these three variables. This would make also any relationship that you find much easier to interpret.

**L. 489 - 491:** This is confusing: In Fig. 8 you try to demonstrate that the $a_{CDOM}275$ - DOC relationship depends on M, here you argue that one regression model is sufficient. Can you elaborate on this (apparent) contradiction?

**Minor aspects (style, wording etc.):**

**L. 12:** Has this algorithm been developed previously? Clarify.

**L. 58:** Replace *acted* by *considered*.

**L. 60 - 61:** Is this not trivial? What other sources exist?

**L. 77:** CDOM is not a single substance.

**L. 110:** What is the particular link between urban water sources and saline lakes?

**L. 168:** Why don't you provide this information for river samples?

**L. 219:** I assume *changed* should be replaced by *ranged*.

**L. 258:** Replace *in* by *during the*.

**L. 599 – 619:** The references are not in the correct alphabetic order.

**L. 643:** The reference is not in the correct alphabetic order.

**Fig. 1:** Please explain what do you mean by *Hydrological features*.

Christian Stamm

**Supplement:**

---

## Referee Comment (RC2) · PK Kowalczuk (Referee) · 17 May 2017

May 17, 2017

Christian Stamm, Ph.D. Associate Editor Hydrology and Earth System Science

Attn: Review of the manuscript by Kaishan Song, Ying Zhao, Zhidan Wen, Chong Fang, and Yingxin Shang entitled "A systematic examination of the relationships between CDOM and DOC in inland waters in China" submitted to Hydrology and Earth System Science and coded hess-2017-179.

Dear Dr Stamm,

[Figure]

After reading the manuscript by Song et al., submitted to Hydrology and Earth System Science and coded hess-2017-179, I think that this study should be consider for publication in this journal after major revision.

General opinion

This study presented results of extensive field studies on relationships between absorption of Chromophoric Dissolved Organic Matter and Dissolved Organic carbon in different water bodies conducted in continental China in different climatic zones. Authors found overall very good correlation between DOC and CDOM absorption coefficient at selected wavelengths, 275 and 400 nm. They have showed that both values of the slope coefficient of the linear regression between considered variables and values of determination coefficient varied considerably between studied water bodies. Author have also proposed a solution to minimize those variations by groping data according to spectral index M, which gave quite uniformed results in respect of the calculated R2, but still there was a significant variability of in regression slope coefficient values. This study proved that application of simple optical measurements could be applied in accurate and reliable estimation of DOC content in fresh water bodies in continental China.

My overall good opinion on this manuscript is somehow hampered by two major flaws: the introduction is overlong with many repetitions especially in regarding remote sensing applications, and Author have written their results together with discussion and it is very difficult for reader to judge when Author presents their own results and when they discuss with published results.

I strongly recommend to reduce introduction to maximum 3-4 pages from current 5, reduce the implications to remote sensing in Introduction. This is particularly redundant because Author have not presented a link between their regression analysis and remote sensing reflectance – the geophysical variable that is physically measured by radiometers placed on spaceborne or airborne platforms. I also strongly recommend

that Author shall present their own results and later give their interpretation in Discussion.

Detailed comments.

Abstract

Page 1 Lines 12 - 13

"An algorithm has been developed to retrieve DOC via CDOM absorption (aCDOM) at 275 and 295 nm for coastal waters, but it is still unclear for the relationship between DOC and aCDOM in other types of waters."

This sentence has no support in presented results. Authors have derived regression relationship between aCDOM(275) and aCDOM(440) but did proposed any remote sensing algorithms in the way it usually developed by the ocean color remote sensing/ocean optics community. Consider to remove this sentence. Abstract shall described your own findings - and shall not contain discussion. When you mention spectral values of aCDOM(ïĄň) – use the symbol l in parenthesis and than indicate specific wavelengths.

Page 2 Lines 28 – 30

Our results indicated the relationships between CDOM and DOC are variable for different inland waters, and therefore remote sensing models for DOC estimation through linking with CDOM absorption need to be tailored according to water types.

This sentence is not precise. Author developed empirical relationships between DOC and aCDOM(ïĄň) but not proposed any remote sensing algorithm. Algorithm need to be developed for different water types and later tested and validated and finally optimized. Please rewrite this sentence. It would be OK in discussion as it points the future direction of your work. Abstract shall briefly and comprehensively present your results.

Introduction

[Figure]

Please reduce length of introduction significantly. Please try to use separate paragraphs to present current knowledge of CDOM biogeochemistry, optics and remote sensing applications to study part of the Earth carbon pool. Just one paragraph thread is sufficient. Avoid later repetitions.

Page 4 Lines 77 – 95

There are a lot overstatements or incorrect sentences in this paragraphs – examples below.

"CDOM is a major light-absorbing substance, which is responsible for much of the color in waters (Reche et al., 1999). "

First of all CDOM is not a substance – it is a heterogeneous mixture of water soluble organic compounds. CDOM have specific optical properties, it absorbs light in UV and visible spectral range and those optical properties change spectral properties and light intensity in water column. From physical point the water color, that can be sensed by human eye (or radiometer) is a ratio between scattering coefficient and sum absorption and scattering coefficients. As CDOM absorption contributes strongly to total absorption coefficient and thus changes the b(ïĄň)/(a(ïĄň)+b(ïĄň)) ratio, the visual effect of CDOM presence in water is change of color to yellowish (or brownish when CDOM concentration is high). That is why first definition of CDOM was "yellow substance".

Page 5 Lines 78 – 80

"The chemical structure and origin of CDOM can be characterized by its absorption coefficients (aCDOM($\lambda$)) and spectral slopes (De Haan and De Boer, 1987; Helms et al., 2008)."

CDOM absorption coefficient aCDOM(ïĄň) cannot characterized CDOM chemical structure – first CDOM is a mixture of countless compounds, second CDOM absorption spectrum is featureless and monotonic and does not contain any spectral peaks that could be associated with specific compounds. Spectral slope of CDOM absorption

spectrum is only an approximate proxy of the relative contribution of fulvic acids and humic acids in this mixture, see Carder et al 1999 for details. There many physical and microbial process influencing effective values of the spectral slope coefficient, so Author shall be caustions using such a definitive statements. All spectral indices cited in following sentences, like SUVA(254), SR etc shall be cited correctly as defined by their Author. Those spectral indices are only optical proxies correlated with sum physical (SR – molecular weight) or chemical (SUVA(254) – relative aromaticity) characteristics of CDOM.

Page 5 Lines 83 – 85

"...while the ratio of CDOM absorption at 250 to 365 nm (aCDOM(250/365), herein, M values) ..."

This ratio shall be defined as aCDOM(250)/aCDOM(365) - not aCDOM(250/365) – this a formal error – please correct throughout the whole manuscript text

"...to track the changes in DOM molecule weight (De Haan and De Boer, 1987; Zhang et al., 2010) and absorption intensity (Song et al., 2013)."

The ratio of two absorption coefficient at two different wavelengths tell nothing about intensity of the absorption process - it only give a relative information who much absorption is stronger(weaker) at one wavelengths relative to other wavelength. Magnitude of ratio by spectral values of absorption coefficients could be an effect of some reasons – according to De Haan and De Boer, 1987 – change in molecular weight). Please cite literature correctly.

Page 5 Lines 91 – 93

"It should be noted that aCDOM(440) is usually used by remote sensing community due to this wavelength is less affected by phytoplankton (Lee et 93 al., 2002)."

This sentence is a complete nonsense. The principle and highest phytoplankton pigments absorption is located at 443 nm. Therefore the effect of phytoplankton absorption on total absorption is highest here. The CDOM absorption in visible range have overlapps with phytoplankton pigments absorption at 443, and this effect was introducing errors in ocean color remote sensing algorithms for retrieval of chlorophyll a concentration. In most cases chlorophyll a was overestimated by those algorithms that were not taking into account CDOM absorption at 443 nm. That was a reason for reporting aCDOM(443) in literature, and inclusion of this parameter particularly in semi0anlytical remote sensing algorithms.

Page 6 Lines 102 – 104

"With compositional change, the absorption feature of CDOM and its relation to DOC varies correspondingly, but the relationship between CDOM and DOC is far from solved (Gonnelli et al., 2013)."

CDOM is a complex mixture of heterogeneous organic compounds, each having individual optical properties. Therefore, the estimation of the universal bulk carbon-specific CDOM absorption coefficient, aâĄŐCDOM($\lambda$), defined as the ratio aCDOM($\lambda$)/DOC, seems almost unfeasible (Woźniak and Dera, 2007). Therefore value of aâĄŐCDOM($\lambda$) may change an order of magnitude in short spatial scale (e.g. Del Vecchio and Blough, 2004; Kowalczuk et al., 2010, Mar Chem 118, 22-36). .

Please consider to rewrite a whole paragraph between lines 77 – 105

Page 6 Line 119

". . . for example the Finish Gulf (Kowalczuk et al., 2006) . . ."

Wrong citation. Paper by Kowalczuk et al., (2006) said nothing about relationship between aCDOM(350) and DOC. This relationship has been presented for Baltic Sea surface waters (not Gulf of Finland) in paper by Kowalczuk et al., (2010) (Oceanologia, 52(3), 431-471). Remove citation to Kowalczuk et al., 2006.

Page 7 Lines 131 – 134

" According to Fig.1, the proposed hypothesis suggests that the main source of . . .."

Repetition. Please try to keep different thread together, do not repeat things that you have said before.

Materials and Methods

Page 9 Line 178

" . . . converted to in situ salinity units (PSU) in the laboratory. "

The salinity in practical salinity scale has no units – it's a ratio of water electrical conductivity measured at given temperature and pressure to ratio of electrical conductivity of artificial sea water measure at standard temperature and pressure. This phrase shall be written as follow:

. . . converted to in situ salinity, expressed in practical salinity scale (PSU), in the laboratory.

Page 9 Line 190

"Chlorophyll-a (Chl-a) was extracted and concentration was measured using a Shimadzu UV-2050PC spectrophotometer (Song et al., 2013)."

Detailed method of spectroscopic measurements of chlorophyll a concentration shall be given, or at least a proper reference to equation that converts measurer absorbance of pigments extract to chlorophyll a concentration shall be cited. Song et al., are not authors of this method, it has been proposed first by Strickland and Parsons, 1972.

Results and discussion

The whole section shall be rewritten to two sections: Results - where Authors presents their own results, and Discussion – where Authors give interpretation of their results.

Page 11 Line 219

Chl-a concentrations (46.44$\pm$59.71 $\mu$g/L) changed from 0.28 to 521.12$\mu$g/L, with the

mean of 46.44 $\mu$g/L.

Redundancy – you give the same value of averaged chlorophyll a concentration twice in the same sentence. Correct.

Page 14 Lines 285 – 287

Phytoplankton degradation may contribute relative large portion of CDOM and DOC in these water bodies (Zhang et al., 2010), due to the lower molecular weight, its absorption is different from that derived from terrestrial systems (Helms et al., 2008).

Wrong citation again. Helms et al., 2008 neither worked in fresh water bodies nor studied phytoplankton degradation products. They have focused on photobleaching effect on spectral slope and have established a spectral slope ratio as proxy for molecular weight. I do not see any information on spectral slope ratio in this paper – so why do you discuss with Helms et al., 2008. This paper does not present any CDOM absorption spectral slope data at all.

The same wrong citation to Helms et al., (2008) repeated on the same page at line 291.

Page 14 – 15, Lines 297 - 300

"As suggested by Brezonik et al. (2015) and Cardille et al. (2013), CDOM in the eutrophic waters or those with very short resident time may show seasonal variation due to algal bloom or hydrological variability, while CDOM in some oligotrhopic lakes or those with long resident time may show an opposite pattern."

This is a part of discussion, but I do not know which part of results is discussed here. Authors did not spent a lot of time on trophic status of studied lakes. The chlorophyll a is mentioned only in one sentence at the beginning of Results section.

Page 15 Line 318

" . . . were found and less colored portion of DOC was presented in waters in semi-arid

to arid regions . . . "

I did not found any data on aCDOM(ïĄň)/DOC relationship in this paper, neither in the text, tables nor figures. What Authors refer to?

Page 16 Line 339

" . . . which is consistent with the findings from Helm et al. (2008) . . ."

Wrong citation again. There is no single line in paper by Helms et al., (2008) on DOC vs. aCDOM(l) relationship.

Page 19 Line 397

" . . . ice and snow cover shielded out most of the solar radiation that might cause a series of biochemical process for CDOM contained in water . . ."

What specific processes Authors refer to? Citation need to support this statements, otherwise I suggest to delete it.

Page 20 Line 428

"This has important implication for remote sensing of DOC through the CDOM absorption as a bridge (Zhu et al., 2014; Kuster et al., 2015; Brezonik et al., 2015)."

What kind of bridge CDOM absorption is ?

Page 23 Line 491

"Most of the paired data sitting close to the regression line except some scattered ones."

Very bizarre sentence that contains no useful information. Delete it.

Conclusion

Delete first two sentences that refer to remote sensing. This paper is about DOC vs. aCDOM(ïĄň) relationships in different water bodies not about remote sensing algorithms.

Page 24 Lines 514 – 516

The slope values of saline lakes and urban waters were close to unity, slope values of river water were highest ($\sim$ 3.1), and slope values of other water types were in between.

Repetition of results – consider to delete.

Acknowledgements

"Last but not the least, the authors 534 would like to thank the editor and two anonymous referees . . .."

Has this manuscript been submitted to other journal and reviewed before current review?

Figure 3 and 5, 8, 9

Y axis legend on figure 3, 5, 8, 9

Is: aCDOM275 (m-1), should be aCDOM(275) [m-1] – please correct accordingly in all specified figures.

Figure 4

Add information to legend – what CDOM absorption coefficient, aCDOM(ïĄň) is presented on 3 panel of Figure 4.

Figure 4

The same remark as for figures 3, 5, 8, 9 – correct Y axis legend to aCDOM(440) [m-1]

Figure 7.

Figure 7 legend the ratio shall be defined as aCDOM(250)/aCDOM(365) - not aCDOM(250/365). Panel a Y axis SUVA(254) dimension is [m2 g-1].

Figure 9

Scales on panel c graph shall be expressed in decimal logarithms log-log. The regression shall be fitted to power function – so it will be linear in log-log scale. See examples in paper by Kowalczuk et al., (2010) (Oceanologia, 52(3), 431-471).

Table 2

Add units to DOC and aCDOM(440) as in Table 1.

Best regards,

Piotr Kowalczuk

Please also note the supplement to this comment:
http://www.hydrol-earth-syst-sci-discuss.net/hess-2017-179/hess-2017-179-RC2-supplement.pdf

---

## Referee Comment (RC3) · Anonymous Referee #2 · 18 May 2017

Piotr Kowalczuk
Institute of Oceanology
Polish Academy of Sciences
Ul. Powstańców Warszawy 55
PL – 81 - 712 Sopot

May 17, 2017

Christian Stamm, Ph.D.
Associate Editor
Hydrology and Earth System Science

Attn: Review of the manuscript by Kaishan Song, Ying Zhao, Zhidan Wen, Chong Fang, and Yingxin Shang entitled "A systematic examination of the relationships between CDOM and DOC in inland waters in China" submitted to Hydrology and Earth System Science and coded hess-2017-179.

Dear Dr Stamm,

After reading the manuscript by Song et al., submitted to Hydrology and Earth System Science and coded hess-2017-179, I think that this study **should be consider** for publication in this journal after **major revision**.

General opinion

This study presented results of extensive field studies on relationships between absorption of Chromophoric Dissolved Organic Matter and Dissolved Organic carbon in different water bodies conducted in continental China in different climatic zones. Authors found overall very good correlation between DOC and CDOM absorption coefficient at selected wavelengths, 275 and 400 nm. They have showed that both values of the slope coefficient of the linear regression between considered variables and values of determination coefficient varied considerably between studied water bodies. Author have also proposed a solution to minimize those variations by groping data according to spectral index M, which gave quite uniformed results in respect of the calculated R2, but still there was a significant variability of in regression slope coefficient values. This study proved that application of simple optical measurements could be applied in accurate and reliable estimation of DOC content in fresh water bodies in continental China.

My overall good opinion on this manuscript is somehow hampered by two major flaws: the introduction is overlong with many repetitions especially in regarding remote sensing applications, and Author have written their results together with discussion and it is very difficult for reader to judge when Author presents their own results and when they discuss with published results.

I strongly recommend to reduce introduction to maximum 3-4 pages from current 5, reduce the implications to remote sensing in Introduction. This is particularly redundant because Author have not presented a link between their regression analysis and remote sensing reflectance – the geophysical variable that is physically measured by radiometers placed on

spaceborne or airborne platforms. I also strongly recommend that Author shall present their own results and later give their interpretation in Discussion.

Detailed comments.

Abstract

Page 1 Lines 12 - 13

"An algorithm has been developed to retrieve DOC via CDOM absorption ($a_{CDOM}$) at 275 and 295 nm for coastal waters, but it is still unclear for the relationship between DOC and $a_{CDOM}$ in other types of waters."

This sentence has no support in presented results. Authors have derived regression relationship between $a_{CDOM}(275)$ and $a_{CDOM}(440)$ but did proposed any remote sensing algorithms in the way it usually developed by the ocean color remote sensing/ocean optics community. Consider to remove this sentence. Abstract shall described your own findings - and shall not contain discussion. When you mention spectral values of $a_{CDOM}(\lambda)$ – use the symbol l in parenthesis and than indicate specific wavelengths.

Page 2 Lines 28 – 30

Our results indicated the relationships between CDOM and DOC are variable for different inland waters, and therefore remote sensing models for DOC estimation through linking with CDOM absorption need to be tailored according to water types.

This sentence is not precise. Author developed empirical relationships between DOC and $a_{CDOM}(\lambda)$ but not proposed any remote sensing algorithm. Algorithm need to be developed for different water types and later tested and validated and finally optimized. Please rewrite this sentence. It would be OK in discussion as it points the future direction of your work. Abstract shall briefly and comprehensively present your results.

Introduction

Please reduce length of introduction significantly. Please try to use separate paragraphs to present current knowledge of CDOM biogeochemistry, optics and remote sensing applications to study part of the Earth carbon pool. Just one paragraph thread is sufficient. Avoid later repetitions.

Page 4 Lines 77 – 95

There are a lot overstatements or incorrect sentences in this paragraphs – examples below.

"CDOM is a major light-absorbing substance, which is responsible for much of the color in waters (Reche et al., 1999). "

First of all CDOM is not a substance – it is a heterogeneous mixture of water soluble organic compounds. CDOM have specific optical properties, it absorbs light in UV and visible spectral range and those optical properties change spectral properties and light intensity in water column. From physical point the water color, that can be sensed by human eye (or

radiometer) is a ratio between scattering coefficient and sum absorption and scattering coefficients. As CDOM absorption contributes strongly to total absorption coefficient and thus changes the $b(\lambda)/(a(\lambda)+b(\lambda))$ ratio, the visual effect of CDOM presence in water is change of color to yellowish (or brownish when CDOM concentration is high). That is why first definition of CDOM was "yellow substance".

Page 5 Lines 78 – 80

"The chemical structure and origin of CDOM can be characterized by its absorption coefficients ($a_{CDOM}(\lambda)$) and spectral slopes (De Haan and De Boer, 1987; Helms et al., 2008)."

CDOM absorption coefficient $a_{CDOM}(\lambda)$ cannot characterized CDOM chemical structure – first CDOM is a mixture of countless compounds, second CDOM absorption spectrum is featureless and monotonic and does not contain any spectral peaks that could be associated with specific compounds. Spectral slope of CDOM absorption spectrum is only an approximate proxy of the relative contribution of fulvic acids and humic acids in this mixture, see Carder et al 1999 for details. There many physical and microbial process influencing effective values of the spectral slope coefficient, so Author shall be caustions using such a definitive statements. All spectral indices cited in following sentences, like SUVA(254), SR etc shall be cited correctly as defined by their Author. Those spectral indices are only optical proxies correlated with sum physical (SR – molecular weight) or chemical (SUVA(254) – relative aromaticity) characteristics of CDOM.

Page 5 Lines 83 – 85

 "…while the ratio of CDOM absorption at 250 to 365 nm (aCDOM(250/365), herein, M values) …"

This ratio shall be defined as $a_{CDOM}(250)/a_{CDOM}(365)$ - not $a_{CDOM}(250/365)$ – this a formal error – please correct throughout the whole manuscript text

"…to track the changes in DOM molecule weight (De Haan and De Boer, 1987; Zhang et al., 2010) and absorption intensity (Song et al., 2013)."

The ratio of two absorption coefficient at two different wavelengths tell nothing about intensity of the absorption process  - it only give a relative information who much absorption is stronger(weaker) at one wavelengths relative to other wavelength. Magnitude  of ratio by spectral values of absorption coefficients could be an effect of  some reasons – according to De Haan and De Boer, 1987 – change in molecular weight). Please cite literature correctly.

Page 5 Lines 91 – 93

"It should be noted that $a_{CDOM}(440)$ is usually used by remote  sensing community due to this wavelength is less affected by phytoplankton (Lee et 93 al., 2002)."

This sentence is a complete nonsense. The principle and highest phytoplankton pigments absorption is located at 443 nm. Therefore the effect of phytoplankton absorption on total absorption is highest here. The CDOM absorption in visible range have overlapps with

phytoplankton pigments absorption at 443, and this effect was introducing errors in ocean color remote sensing algorithms for retrieval of chlorophyll a concentration. In most cases chlorophyll a was overestimated by those algorithms that were not taking into account CDOM absorption at 443 nm. That was a reason for reporting $a_{CDOM}(443)$ in literature, and inclusion of this parameter particularly in semi0anlytical remote sensing algorithms.

Page 6 Lines 102 – 104

"With compositional change, the absorption feature of CDOM and its relation to DOC varies correspondingly, but the relationship between CDOM and DOC is far from solved (Gonnelli et al., 2013)."

CDOM is a complex mixture of heterogeneous organic compounds, each having individual optical properties. Therefore, the estimation of the universal bulk carbon-specific CDOM absorption coefficient, $a^*_{CDOM}(\lambda)$, defined as the ratio $a_{CDOM}(\lambda)/DOC$, seems almost unfeasible (Woźniak and Dera, 2007). Therefore value of $a^*_{CDOM}(\lambda)$ may change an order of magnitude in short spatial scale (e.g. Del Vecchio and Blough, 2004; Kowalczuk et al., 2010, Mar Chem 118, 22-36). .

Please consider to rewrite a whole paragraph between lines 77 – 105

Page 6 Line 119

"… for example the Finish Gulf (Kowalczuk et al., 2006) …"

Wrong citation. Paper by Kowalczuk et al., (2006) said nothing about relationship between $a_{CDOM}(350)$ and DOC. This relationship has been presented for Baltic Sea surface waters (not Gulf of Finland) in paper by Kowalczuk et al., (2010) (Oceanologia, 52(3), 431-471). Remove citation to Kowalczuk et al., 2006.

Page 7 Lines 131 – 134

" According to Fig.1, the proposed hypothesis suggests that the main source of …."

Repetition. Please try to keep different thread together, do not repeat things that you have said before.

Materials and Methods

Page 9 Line 178

" … converted to in situ salinity units (PSU) in the laboratory. "

The salinity in practical salinity scale has no units – it's a ratio of water electrical conductivity measured at given temperature and pressure to ratio of electrical conductivity of artificial sea water measure at standard temperature and pressure. This phrase shall be written as follow:

… converted to in situ salinity, expressed in practical salinity scale (PSU), in the laboratory.

Page 9 Line 190

"Chlorophyll-a (Chl-a) was extracted and concentration was measured using a Shimadzu UV-2050PC spectrophotometer (Song et al., 2013)."

Detailed method of spectroscopic measurements of chlorophyll a concentration shall be given, or at least a proper reference to equation that converts measurer absorbance of pigments extract to chlorophyll a concentration shall be cited. Song et al., are not authors of this method, it has been proposed first by Strickland and Parsons, 1972.

Results and discussion

The whole section shall be rewritten to two sections: Results - where Authors presents their own results, and Discussion – where Authors give interpretation of their results.

Page 11 Line 219

Chl-a concentrations (46.44±59.71 µg/L) changed from 0.28 to 521.12µg/L, with the mean of 46.44 µg/L.

Redundancy – you give the same value of averaged chlorophyll a concentration twice in the same sentence. Correct.

Page 14 Lines 285 – 287

Phytoplankton degradation may contribute relative large portion of CDOM and DOC in these water bodies (Zhang et al., 2010), due to the lower molecular weight, its absorption is different from that derived from terrestrial systems (Helms et al., 2008).

Wrong citation again. Helms et al., 2008 neither worked in fresh water bodies nor studied phytoplankton degradation products. They have focused on photobleaching effect on spectral slope and have established a spectral slope ratio as proxy for molecular weight. I do not see any information on spectral slope ratio in this paper – so why do you discuss with Helms et al., 2008. This paper does not present any CDOM absorption spectral slope data at all.

The same wrong citation to Helms et al., (2008) repeated on the same page at line 291.

Page 14 – 15, Lines 297 - 300

"As suggested by Brezonik et al. (2015) and Cardille et al. (2013), CDOM in the eutrophic waters or those with very short resident time may show seasonal variation due to algal bloom or hydrological variability, while CDOM in some oligotrhopic lakes or those with long resident time may show an opposite pattern."

This is a part of discussion, but I do not know which part of results is discussed here. Authors did not spent a lot of time on trophic status of studied lakes. The chlorophyll a is mentioned only in one sentence at the beginning of Results section.

Page 15 Line 318

" …  were found and less colored  portion of DOC was presented in waters in semi-arid to arid regions … "

I did not found any data on $a_{CDOM}(\lambda)$/DOC relationship in this paper, neither in the text, tables nor figures. What Authors refer to?

Page 16 Line 339

" … which is consistent with the findings from Helm et al. (2008) …"

Wrong citation again. There is no single line in paper by Helms et al., (2008) on DOC vs. $a_{CDOM}(l)$ relationship.

Page 19 Line 397

" … ice and snow cover shielded out most of the solar radiation that  might cause a series of biochemical process for CDOM contained in water …"

What specific processes Authors refer to? Citation need to support this statements, otherwise I suggest to delete it.

Page 20 Line 428

"This has important  implication for remote sensing of DOC through the CDOM absorption as a bridge  (Zhu et al., 2014; Kuster et al., 2015; Brezonik et al., 2015)."

What kind of bridge CDOM absorption is ?

Page 23 Line 491

"Most of the paired data sitting close to the regression  line except some scattered ones."

Very bizarre sentence that contains no useful information. Delete it.

Conclusion

Delete first two sentences that refer to remote sensing. This paper is about DOC vs. $a_{CDOM}(\lambda)$ relationships in different water bodies not about remote sensing algorithms.

Page 24 Lines 514 – 516

The slope values of saline lakes and  urban waters were close to unity, slope values of river water were highest (~ 3.1), and slope values of other water types were in between.

Repetition of results – consider to delete.

Acknowledgements

"Last but not the least, the authors 534 would like to thank the editor and two anonymous referees ….."

Has this manuscript been submitted to other journal and reviewed before current review?

Figure 3 and 5, 8, 9

Y axis legend on figure 3, 5, 8, 9

Is: $a_{CDOM}275$ (m-1), should be $a_{CDOM}(275)$ $[m^{-1}]$ – please correct accordingly in all specified figures.

Figure 4

Add information to legend – what CDOM absorption coefficient, $a_{CDOM}(\lambda)$ is presented on 3 panel of Figure 4.

Figure 4

The same remark as for figures 3, 5, 8, 9 – correct Y axis legend to $a_{CDOM}(440)$ $[m^{-1}]$

Figure 7.

Figure 7 legend the ratio shall be defined as $a_{CDOM}(250)/a_{CDOM}(365)$ - not $a_{CDOM}(250/365)$. Panel a Y axis SUVA(254) dimension is $[m^2\,g^{-1}]$.

Figure 9

Scales on panel c graph shall be expressed in decimal logarithms log-log. The regression shall be fitted to power function – so it will be linear in log-log scale. See examples in paper by Kowalczuk et al., (2010) (Oceanologia, 52(3), 431-471).

Table 2

Add units to DOC and $a_{CDOM}(440)$ as in Table 1.

Best regards,

Piotr Kowalczuk

---

## Author Comment (AC1) · 24 Jun 2017

This paper presents a series of regression equations between DOC concentrations and optical properties of the DOM across a range of water bodies in China. The authors have amassed an impressive data set, and applying this data set to questions of DOC biogeochemistry could make a useful contribution. Unfortunately the paper, as

currently written, has some flaws that limit its value.

Responses: The authors thank for the positive comments on the impressive dataset, and also pointing out the flaws listed below, which we have addressed in detail after each comment or suggestion forwarded be the reviewer.

The paper focuses on two objectives stated in the Introduction, and a third objective that, for some reason, is presented in the Methods (lines 157-159). The objectives all are targeted at examining the relationship between DOC concentrations and optical properties, particularly absorbance at275nm or 440nm. The paper would be improved if it were structured around testable hypotheses, which I think the authors could do without too much additional work.

Responses: The authors really thank for the reviewer's instructive comments, we added testable hypotheses in the revised manuscript, and structured the layout of the manuscript according to the testable hypotheses. Thanks again for the valuable comments that really help for the improvement of the manuscript.

The primary means of data analysis is simple linear regression, and it appears that perhaps multiple linear regression was attempted (line 279-280). Surprisingly, no description of data analysis is provided in the paper (or the supplemental information). In fact, P values are not even provided for the regression analyses. Nor is there any indication of testing for normality or other assumptions for linear regression.

Responses: The authors thank for the comments. In the revised manuscript, descriptive statistical analysis were conducted for the data set, and assumptions for the linear regression were also tested for these regression analysis, in addition, P values for each regression model also were also provided in the revised manuscript. Many of the graphs show that a single data point, or a couple data points, appears to be leveraging the relationship (e.g., Fig 3c, Fig 3e, Fig 3f, Fig 6d, Fig 6f, and others). In these cases, the validity of the regression equation is highly questionable.

Responses: The authors thank for the comments. We agree that a single data point or a couple data points might have improved the R-squares for these regression models, however, these data points are in situ measured values, and thus they reflect the natural situation. In the revised manuscript, we also did the regressions without these data points, and the results indicated that the R-squares did not affected much. We really appreciated your thoughtful comments. We could provide these regression metrics with and without these points in the revised manuscript.

In the case of Fig 8, it is not clear how the groupings were selected. The text mentions "trial and error" which suggests to be it was a very subjective process of selecting the M ranges for the groups. Responses: The authors thank for the comments. In the current manuscript, the results presented in Figure 8 were derived based on trial and error testing of the regression modeling. The M value is used to classify CDOM into different groups, which might have similar CDOM absorption efficiency or absorption ability in each group, thus the CDOM absorption coefficient in each group should have similar relationship with DOC. However, how to determine the range for each group is still very subjective, we will further investigate and try to find a more reliable method for the grouping process. The testing results will be presented in the revised manuscript, thanks again for the comments. I am a bit concerned about the holding time (up to 2 days) before fiAltration. Do the authors have any evidence that there was no degradation of DOC during the holding time? Some concern for chlorophyll-a. Also, it is questionable to collect and store DOC for optical analysis in HDPE bottles. Why was HDPE used instead of glass?

Responses: The authors thank for the very thoughtful comments. All the water samples ship back to laboratory and then stored in refrigerator at about 4℃ in the dark, thus the biodegradation should be very limited for DOC at low temperature. Similarly, the photo-degradation is also avoided since samples were kept in the dark. Some literatures also addressed this issue, and found that DOC is relatively stable, its change in two days at low temperature without photo-degradation should be neglectable. As

for the HDPE sampling bottle, according to my knowledge, it is quite common to use HDPE bottles for field sampling to test water quality parameters, which has been previously cleaned by soaking in 0.5 mol LÂň-1 HCl followed by 0.1 mol L-1NaOH for 24 h before heading to the field. According to Zhang et al. (2007), samples kept in two day before filtering would not cause obvious degradation for DOC concentration. Using glass bottle is not easy to ship back from field to laboratory during the bad road conditions, especially in Tibet or other remote areas where county roads are very common, which could cause severe damage of the glass bottles, thus HDPE bottles were used.

In the end, the authors state that SUVA is not an appropriate metric for the purposes of their study because its calculation includes DOC concentration. This left me wondering why it was included at all?

Responses: The authors thank you for the comments. Actually, we used both SUVA and M value (a250/a365) to characterize CDOM molecular weight qualitatively, and particularly SUVA is a very effective index for characterizing the molecular size of CDOM, thus we prefer the keep this part in the manuscript, but its linkage with CDOM grouping will be removed in the revised manuscript. Thanks again for your kind concern.

I think the Introduction could be shortened by as much as a third without any loss. Much of the introduction deals with remote sensing for DOC, but this paper does not address remote sensing directly; the background information on remote sensing could be greatly reduced within the Introduction and also the Discussion. I think developing some testable hypotheses and keeping the Introduction (and the whole paper) focused narrowly on those hypotheses would make for a shorter, and more readable, paper.

Responses: The authors really thank for the reviewer's very instructive comments. As you may see that there is another reviewer who also suggests to shorten this part, thus, the Introduction will be shortened in the revised manuscript.

I would strongly suggest separate Results and Discussion sections. As I read the paper

it was not always clear when the authors were making statements based on their data, versus general statements from literature.

Responses: Again, the authors thank for the very thoughtful comments, and similar comment were also raised by the third reviewer (Professor P.K.Kowalczuk), we separate the Results and Discussion sections in the revised manuscript. We really appreciate this comments, which would definitely strengthen this manuscript.

Try to avoid vague statements such as "massive organic matter" (line 22) and "big variation" (line 230). The English in the paper is mostly correct, but it could certainly be improved if edited closely by a native English speaker.

Responses: The authors thank for the comments, your kind comments were adapted in the revised manuscript, further, and the authors have requested Professor Lin Li from IUPUI (Indiana University Purdue University, Indianapolis) edit the English in the revised manuscript.

---

## Author Comment (AC2) · 24 Jun 2017

The Authors mention in the text that using remote sensing for small ponds and lakes is problematic because of lack of appropriate remote sensing sensors. This may have been true some time ago (Palmer et al. 2015), but is not any more. Sentinel-2 imagery with 10 m spatial resolution is available globally. This kind of resolution is suitable for almost any pond, not speaking about lakes. Sentinel-2A data has been already used in mapping lake CDOM and DOC (Toming et al. 2016). Sentinel-2B was launched two months ago and is currently in testing phase. Meaning that in a few months 10 m spatial resolution imagery will be available with 5 days revisit time at the equator and

about 2-3 day revisit time for most lakes in China. Besides that Landsat-8 imagery with 30 m spatial resolution is also available. There have been several papers recently showing the usefulness of Landsat-8 in mapping lake CDOM/DOC. Consequently, the image data is not a problem anymore. This strengthens the value of this research even more. I recommend to improve the remote sensing part of the manuscript showing that there is plenty of data available now free of charge with very high spatial and temporal resolution and your study will help to improve usefulness of this data at very local to global scales.

Response: the authors really thank Professor Kutser's valuable and very instructive comments. These valuable comments will be definitely helpful in revising the current manuscript, and the manuscript in preparation, which is mainly focused on establishing an algorithm with remotely sensed imagery data (e.g., Landsat OLI,Sentinel-2A, and Sentinel-3A/OLCI). For the current manuscript, the major objective is to examine the variation for the relationship between DOC and aCDOM($\lambda$i), which has the potential to be applied for DOC estimate in inland waters. As stated in the introduction section of manuscript, the regression model slopes may vary significantly for different water types that ultimately affect DOC estimated results. Thus, we mainly focus on the relationship between DOC and CDOM absorptions for different types of waters. As you may see that the two other reviewers both suggested to remove the remote sensing part since no algorithm were established specifically for each types of waters being concerned in this study. As aforementioned, your kind suggestions will definitely be incorporated in the manuscript in preparation, which is mainly focused on remote estimate of DOC concentration through the relationship between CDOM and DOC tracked in this study based on the optical classification of different types of waters. Thanks again for the very instructive comments.

SUVA is an important parameter used to describe carbon quality (e.g. in drinking water industry). Therefore, it is important to link remote sensing and SUVA more closely in the manuscript. Remote sensing of SUVA has been demonstrated at least in one recent

paper cited several times by the Authors. I would recommend to add this reference in the 3.4 and strengthen the link between SUVA and remote sensing there.

Response: the authors really thank for the suggestions. Same like the responses to the exactly previous comments, the authors will retain the current manuscript major topic, and only focus on the relationship between DOC and CDOM, and the remote sensing part will be addressed in the manuscript in preparation. Thus, all you kind suggestions will be definitely incorporated in that manuscript, hope you could give more instructive comments on the ongoing one later on. Is there any information available for seasonal variability? At least in boreal zone CDOM decreases from spring to summer and then starts to increase again(e.g. Kutser 2012), but how about the CDOM-DOC or DOC-SUVA relationships? This would be a very interesting piece of information.

Response:thanks for the valuable comments, certainly, the attempts to examine the temporal variability between DOC and CDOM would be very interesting piece of information, however, there only one visit for most of the waters being sampled. But, we have water samples collected in three river sections in weekly or bi-weekly time steps, which indicated that CDOM-DOC relationship (see Figure 5) may change with different rivers. The head water section shows higher regression slope, while river with certain amount of anthropogenic pollution will result in decreased regression slope value (Figure 5c, sample were collected in the Songhua River, which was polluted by sewage waters and other anthropogenic sources).

In general the paper is written well. There are some minor errors in names (e.g. must be Gulf of Finland not Finish Gulf in row 119) and some sentences can be modified, but the text is easily readable.

Response: the authors really thank for the valuable comments, these minor errors and some of the problematic sentences were corrected or rephrased in the revised manuscript, thanks again for the positive comments.

179, 2017.

---

## Author Comment (AC3) · 24 Jun 2017

C. Stamm (Editor) christian.stamm@eawag.ch

Editor comment HESSD-Manuscript "A systematic examination of the relationships between CDOM and DOC in inland waters in China" (HESS 2017-179).

Dear Dr. K. Song

Reading through the manuscript, I came across a number of aspects on which I'd like

to comment on during this discussion phase.

Comments on content: L.70-75: Provide explanations on mechanisms how hydrology affects DOC and CDOM properties. Why shall catchment size per se be important? Can you explain why size is influential apart from affecting for example travel times?

Response: The authors really thank for your thoughtful comments. To my knowledge, these hydrological factors may influence DOC and CDOM properties, 1) the source of DOC and CDOM drain to rivers from the catchment, thus the landscape and the soil organic density influence DOC and CDOM abundance in the rivers; 2) the hydrograph is another factor influences DOC concentration in rivers, generally before peak flow the river shows relatively lower DOC, but relatively higher DOC exhibits after the peak flow due to more DOC release from soil; 3) generally small river catchment tends to have homogeneous landscape, and DOC and CDOM easily drain to river without too much change during this draining processes, that explains why head water generally exhibit higher DOC and CDOM, also more close relationship reveals for DOC and CDOM in head waters; in terms of larger catchment, larger variability for landscape, soil properties will exhibit, longer travel time takes place (photo-degradation and microbial degradation reduce the colored fraction of DOC, and also DOC will mineralize), which ultimately affect the CDOM and DOC properties. Also, rivers in tropical or subtropical regions tend to show lower DOC, which is mainly due to the higher frequency of flushing dilutes the DOC in rivers; comparatively, less rainfall produce less surface flow in temperature regions, where higher DOC generally exhibit in rivers in these region, of course relatively higher soil organic matter also contributes the higher DOC and CDOM in temperature rivers.

L.316: If you know about these influencing factors, why you cannot derive an explanatory model for the slope?

Response: Thanks for your comments, but I am not sure I fully understand your comments. We roughly know these influencing factors, however, the variation caused by

each factors and the contribution to the total variation by each factor are not clear, and also these factors are intermingled or interacted each other, thus it is very hard to establish an explanatory model for the slope. In addition, the relationship between DOC and CDOM for different waters varies substantially due to the compositional differences for CDOM, and the fraction of colored components in DOC is changeable, which ultimately influences the relationship between DOC and CDOM, thus only a relative stable regression model is achievable for a specific types of waters, not possible for all types of waters. I am not sure if I have answered your question.

L.345-347: Again, you should explain how hydrology and catchment characteristics can influence CDOM and DOC.

Response: Thanks for your valuable comments. Most of the explanation was presented in the responses above. Further, I would highlight that rivers in arid or semi-arid regions (through our work and also work by Spencer et al. (2012, JGR)) generally exhibit higher DOC concentration, but the absorption coefficient for CDOM is generally low with higher spectral slope, in which the high concentration of DOC is caused by the condensed effect through evaporation;as for the deep spectral slope, the longer traveling time, strong dose of irradiance with less cloud cover, can be attributed for this phenomenon.

L.350-357,Fig. 4: This part is highly misleading because the text evokes the impression that you compare DOC and CDOM to simultaneous measurements of discharge. Unfortunately, this is not the case. You write in the manuscript that the hydrographs correspond to long term averages and do not represent the actual flow conditions during the periods of your sampling campaign. However, you do notpayattentiontothatbasic-factwhendisplayingthedata: Byplottingdischarge and concentrations against the same time axis (Fig. 4) you give the impression that a flow rate value on day X is linked to the concentrations for the same day. However, this is not true. Therefore, this way of presenting and interpreting the data is not acceptable. Such figures would only be correct if you can provide the discharge data from the years of sampling. Should it not

be possible to get access to this data, you have to adapt you data analysis accordingly. Because you neither have information about the actual flow for a given sampling date nor about the actual sequence of actual discharge from day to day you should not plot the data against the time axis. Instead you plot for example the concentrations against a selected quantile of flow rates for the corresponding Julian day. This would also be more to the point because you argue that there is a relationship between flow rate and DOC/CDOM concentrations irrespective of the Julian day.

Response: Thanks for your comments, I am really sorry for this misunderstanding which caused by my carelessness for the "note" in the caption for Figure 4 was not removed, thus the previous caption was not changed according to the updated flow data. In the previous version, the hydrograph is the average value, however, the in the current version, we tried the best and purchase the concurrent river flow data (we could provide these data in excel if necessary), so all the flow data are concurrent to the sampling year. I am really sorry for this misunderstanding, which is my fault not updating the caption in Figure 4 conveying the wrong information.

L.425-426,Fig. 3,Fig. 6: Thedatasetsinthetwofiguresseemnotbefullyconsistent. When comparing for example Fig. 3d and 6d, there are $3-4$ data points with DOC concentrations of $80-120$ mg L$-1$ in Fig. 3d that are absent in Fig. 6d. How does it come? The same holds in the opposite direction with data of about 45 mg L$-1$ in Fig. 6c. What is the explanation?

Response: Thanks for your comments that really help to improve the manuscript. As you may know that the manuscript has revised at least two times, and some of the figures were updated, removed or added. The first version of the data processed by myself, and the figure 6 was provided in the last revised version, in which the data was processed by my students. The large data set were collected in different field campaigns, also some data were added in the revised manuscript, thus, inconsistency was caused due to our change of data processing person, and new data incorporated in the revised manuscript. We will further check the data set, and make sure all the

data sets are consistent. Thanks again for pointing out the inconsistency, that really help for improving our manuscript, also we should bear in mind for being more careful during data processing and manuscript preparing.

L.451: I cannot see this low M values in Fig. 7B. Can you support your statement by a statistical metric?

Responses:the authors thank for the comments, we could provide the statistical metric in the revised manuscript.

L.462: Your selection of categories for M is rather arbitrary. Additionally, when looking at them in their entirety, it is obvious that there is a general pattern in that the slope decreases with increasing M (see figure in the attachment). Hence, the slope simultaneously depends on DOC and M. Because M is simply the ratio between aCDOM250 and aCDOM365, it follows that aCDOM275 is a function of DOC, aCDOM250, and aCDOM365. Instead of introducing first M and then classify M into categories, you better directly express aCDOM275 as a function of these three variables. This would make also any relationship that you find much easier to interpret.

Response: The authors really thank you for the thoughtful comments. Here, I might not make myself clear, we tried stepwise regression with CDOM absorption at 250 nm, 275nm, and 365 nm according to your kind suggestion, and found that there is no significant improvement for DOC estimate. It is two different things by incorporating M into regression model, and by grouping CDOM into different groups based on M value. In fact, if we try to incorporate aCDOM250 and aCDOM365 into the regression model, these two variables are just absorption intensity that won't help two much for the regression model. The essential thing here is to group CDOM of different waters into various group based on M values, thus each group roughly have similar absorption characteristics, which ultimately helps to improve the model accuracy. However, if incorporate M into regression model, which won't help for the accuracy.

L.489-491: This is confusing: In Fig. 8 you try to demonstrate that the aCDOM275 -

DOC relationship depends on M, here you argue that one regression model is suffi-cient. Can you elaborate on this (apparent) contradiction?

Response: the authors thank you for the concern. As you might have noticed that these few high values of DOC and CDOM have leveraged the good relationship (Figure 9a), and remove these points will decrease the relationship between DOC and CDOM. Further, the DOC concentrations are still very high in Figure 9b, however the R-square value is only about 0.66. Thus, this model is roughly accurate for inland waters at national or sub-continental scale with large variation of DOC and CDOM. However, if accurate estimate of DOC through CDOM absorption needs to be achieved, then different types of waters should be classified, for instance based on M values which help to differentiate CDOM absorption efficiency. Further, the regression based on pooled data also give readers an approximate idea how the relationship between DOC and CDOM looks like with large data set covering different types of inland waters (river, freshwater, saline water, and urban water influenced by effluent and sewage discharge). If you suggest to remove this part, we would be happy to do so, thanks again for your consideration.

Minor aspects (style, wording etc.):

L.12: Has this algorithm been developed previously? Clarify.

Response: this algorithm was developed by Fichot and Benner in 2011, and it was clarified in the revised manuscript, thanks for the comments.

L.58: Replace acted by considered. Response: the authors thank for the suggestion, we replaced 'acted' with 'considered'.

L.60-61: Is this not trivial? What other sources exist?

Response: the authors thank for the comment, the sentence was rephrased. The sources are external, internal, and anthropogenic origin.

L.77: CDOM is not a single substance.

Response: the authors thank for the comment, we replaced 'substance' with 'constituent' in the revised manuscript.

L.110: What is the particular link between urban water sources and saline lakes? Response:sorry for the ambiguous statement, there is no link between urban water sources and saline lakes, the authors rephrased this sentence in the revised manuscript.

L.168: Why don't you provide this information for river samples?

Response: the authors thank for the concern, I might have clarified in the responses for the previous review and the resubmission for the current manuscript, we sampled quite a lot river sections, and some of these rivers or streams even don't have name for it, particularly for these in Tibet Plateau, thus it is not practical for use to provide this information. We could provide these sampling station with a map if necessary.

L.219: I assume changed should be replaced by ranged.

Response: The authors really thank for the comment, your kind suggestion was incorporated in the revised manuscript.

L.258: Replace in by during the. Response: your kind suggestion was incorporated in the revised manuscript, Thanks a lot for comment.

L.599–619: The references are not in the correct alphabetic order.

Response: The authors really thank for comment, the correct alphabetic order for the references were achieved in the revised manuscript.

L.643: The reference is not in the correct alphabetic order.

Response: The authors really thank for comment, the right order was corrected.

Fig. 1: Please explain what do you mean by Hydrological features.

Response: The authors thank for the concern, here the authors mainly talk about the

lake morphologic characteristics. We could add note in the figure caption to explain what the hydrological features are in this context.

---

## Author Comment (AC4) · 24 Jun 2017

PK Kowalczuk (Referee) piotr@iopan.gda.pl

May 17, 2017 Christian Stamm, Ph.D. Associate Editor Hydrology and Earth System Science

Attn: ReviewofthemanuscriptbyKaishanSong,YingZhao,ZhidanWen,ChongFang, and Yingxin Shang entitled "A systematic examination of the relationships between CDOM

and DOC in inland waters in China" submitted to Hydrology and Earth System Science and coded hess-2017-179.

Dear Dr Stamm,

After reading the manuscript by Song et al., submitted to Hydrology and Earth System Science and coded hess-2017-179, I think that this study should be consider for publication in this journal after major revision.

General opinion

This study presented results of extensive field studies on relationships between absorption of Chromophoric Dissolved Organic Matter and Dissolved Organic carbon in different water bodies conducted in continental China in different climatic zones. Authors found overall very good correlation between DOC and CDOM absorption coefficient at selected wavelengths, 275 and 400 nm. They have showed that both values of the slope coefficient of the linear regression between considered variables and values of determination coefficient varied considerably between studied water bodies. Author have also proposed a solution to minimize those variations by groping data according to spectral index M, which gave quite uniformed results in respect of the calculated R2, but still there was a significant variability of in regression slope coefficient values. This study proved that application of simple optical measurements could be applied in accurate and reliable estimation of DOC content in fresh water bodies in continental China.

My overall good opinion on this manuscript is somehow hampered by two major flaws: the introduction is overlong with many repetitions especially in regarding remote sensing applications, and Author have written their results together with discussion and it is very difficult for reader to judge when Author presents their own results and when they discuss with published results.

I strongly recommend to reduce introduction to maximum 3-4 pages from current 5,

reduce the implications to remote sensing in Introduction. This is particularly redundant because Author have not presented a link between their regression analysis and remote sensing reflectance – the geophysical variable that is physically measured by radiometers placed on spaceborne or airborne platforms. I also strongly recommendthat Author shall present their own results and later give their interpretation in Discussion.

Responses: The authors thank for the positive comments on the overall quality of the manuscript, particularly for the data set. Also, the authors thank for Professor Kowalczuk pointing out the two major flaws, which we have addressed in the revised manuscript by shortening or removing some unnecessary parts relevant to remote sensing application in the Introduction section; further, we will separate Results section with Discussion section in the revised manuscript.

Detailed comments.

Abstract

Page 1 Lines 12 – 13 "An algorithm has been developed to retrieve DOC via CDOM absorption (aCDOM) at 275 and 295 nm for coastal waters, but it is still unclear for the relationship between DOC and aCDOM in other types of waters." Thissentencehasnosupportinpresentedresults.AuthorshavederivedregressionrelationshipbetweenaCDOM(275)andaCDOM(440)butdidproposedanyremotesensi optics community. Consider to remove this sentence. Abstract shall described your own findings - and shall not contain discussion. When you mention spectral values of aCDOM(ïAËŻËĞn) – use the symbol l in parenthesis and then indicate specific wavelengths.

Responses: The authors thank for the instructive and specific comments, we removed this sentence in the revised manuscript. The very instructive comment for presentation of aCDOM by including specific wavelengths was incorporated throughout the manuscript during the revision.

Page 2 Lines 28 – 30 Our results indicated the relationships between CDOM and DOC are variable for different inland waters, and therefore remote sensing models for DOC estimation through linking with CDOM absorption need to be tailored according to water types. This sentence is not precise. Author developed empirical relationships between DOC and aCDOM(ïAËŻËĞn) but not proposed any remote sensing algorithm. Algorithm need to be developed for different water types and later tested and validated and finally optimized. Please rewrite this sentence. It would be OK in discussion as it points the future direction of your work. Abstract shall briefly and comprehensively present your results.

Responses: The authors thank for the thoughtful comments, we rewrote this sentence in the revised manuscript to avoid misunderstanding with remote sensing of DOC through the linkage with CDOM, we will try the best to achieve a concise and comprehensive abstract in the revised manuscript.

Introduction Please reduce length of introduction significantly. Please try to use separate paragraphs to present current knowledge of CDOM biogeochemistry, optics and remote sensing applications to study part of the Earth carbon pool. Just one paragraph thread is sufficient. Avoid later repetitions.

Responses: The authors thank for the comments, the Introduction section was separated into current knowledge of CDOM biogeochemistry, optics and remote sensing applications. Thanks again for the suggestions that really make the Introduction presented more logically.

Page 4 Lines 77 – 95 There are a lot overstatements or incorrect sentences in this paragraphs – examples below. "CDOMisamajorlight-absorbingsubstance,whichisresponsibleformuchofthecolor in waters (Reche et al., 1999). " First of all CDOM is not a substance – it is a heterogeneous mixture of water soluble organic compounds. CDOM have specific optical properties, it absorbs light in UV and visible spectral range and those optical properties change

spectral properties and light intensity in water column. From physical point the water color, that can be sensed by human eye (orradiometer) is a ratio between scattering coefficient and sum absorption and scattering coefficients. As CDOM absorption contributes strongly to total absorption coefficient and thus changes the $b(\lambda)/(a(\lambda)+b(\lambda))$ ratio, the visual effect of CDOM presence in water is change of color to yellowish (or brownish when CDOM concentration is high). That is why first definition of CDOM was "yellow substance". Responses: The authors thank for the very detailed comments, which really help for clarifying the role that CDOM plays in water color remote sensing, or the water leaving radiance by optically active constituents. Your kind suggestions were absorbed and incorporated in the revised manuscript, and some of the inappropriate statements were rephrased.

Page 5 Lines 78 – 80 "The chemical structure and origin of CDOM can be characterized by its absorption coefficients (aCDOM($\lambda$)) and spectral slopes (De Haan and De Boer, 1987; Helms et al., 2008)."

CDOM absorption coefficient aCDOM($\lambda$) cannot characterized CDOM chemical structure – first CDOM is a mixture of countless compounds, second CDOM absorption spectrum is featureless and monotonic and does not contain any spectral peaks that could be associated with specific compounds. Spectral slope of CDOM absorption spectrum is only an approximate proxy of the relative contribution of fulvic acids and humic acids in this mixture, see Carder et al 1999 for details. There many physical and microbial process influencing effective values of the spectral slope coefficient, so Author shall be cautious using such a definitive statements. All spectral indices cited in following sentences, like SUVA(254), SR etc shall be cited correctly as defined by their Author. Those spectral indices are only optical proxies correlated with sum physical (SR – molecular weight) or chemical (SUVA(254) – relative aromaticity) characteristics of CDOM.

Responses: The authors really thank for the reviewer's very instructive comments. These helpful suggestions or comments were adopted in the revised manuscript.

[Figure]

Page 5 Lines 83 – 85 "...while the ratio of CDOM absorption at 250 to 365 nm (aCDOM(250/365), herein, M values) ..." ThisratioshallbedefinedasaCDOM(250)/aCDOM(365)-notaCDOM(250/365)–this a formal error – please correct throughout the whole manuscript text.

Responses: The authors thank for the instructive comments. The authors replaced "aCDOM(250/365)" with "aCDOM(250)/aCDOM(365)" throughout the revised manuscript.

"...to track the changes in DOM molecule weight (De Haan and De Boer, 1987; Zhang et al., 2010) and absorption intensity (Song et al., 2013)." The ratio of two absorption coefiňĄcient at two different wavelengths tell nothing about intensity of the absorption process-it only give a relative information who much absorption is stronger(weaker) at one wavelengths relative to other wavelength. Magnitude of ratio by spectral values of absorption coefiňĄcients could be an effect of some reasons – according to De Haan and De Boer, 1987 – change in molecular weight). Please cite literature correctly.

Responses: The authors really thank for the reviewer's very instructive comments. The right citations were provide in the revised manuscript.

Page 5 Lines 91 – 93 "It should be noted that aCDOM(440) is usually used by remote sensing community due to this wavelength is less affected by phytoplankton (Lee et al., 2002)." This sentence is a complete nonsense. The principle and highest phytoplankton pigments absorption is located at 443 nm. Therefore the effect of phytoplankton absorption on total absorption is highest here. The CDOM absorption in visible range have overlapps with phytoplankton pigments absorption at 443, and this effect was introducing errors in ocean color remote sensing algorithms for retrieval of chlorophyll a concentration. In most cases chlorophyll a was overestimated by those algorithms that were not taking into account CDOM absorption at 443 nm. That was a reason for reporting aCDOM(443) in literature, and inclusion of this parameter particularly in semi0anlytical remote sensing algorithms.

Responses: This comments is very instructive, that really help me understand the underlying reason why aCDOM(443) was reported in remote sensing community. Thanks again for the reviewer's valuable comments.

Page 6 Lines 102 – 104 "With compositional change, the absorption feature of CDOM and its relation to DOC variescorrespondingly,buttherelationshipbetweenCDOMandDOCisfarfromsolved (Gonnelli et al., 2013)."

CDOM is a complex mixture of heterogeneous organic compounds, each having individual optical properties. Therefore, the estimation of the universal bulk carbon-specific CDOM absorption coefficient, aâAËŻËİOCDOM($\lambda$), defined as the ratio aCDOM($\lambda$)/DOC,seemsalmostunfeasible(Woz′niakandDera,2007). Thereforevalue of aâAËŻËİOCDOM($\lambda$) may change an order of magnitude in short spatial scale (e.g. Del Vecchio and Blough, 2004; Kowalczuk et al., 2010, Mar Chem 118, 22-36).

Responses: The authors really thank for the instructive comments, which has been incorporated in the revised manuscript, and these references recommended by the reviewer were also adopted during the manuscript revision.

Please consider to rewrite a whole paragraph between lines 77 – 105

Responses: Again, the authors thank for the comment, and the whole paragraph was rewritten in the revised manuscript, and all the reviewer's comments for the whole paragraph listed above were also incorporated during rewriting of this part.

Page 6 Line 119 "... for example the Finish Gulf (Kowalczuk et al., 2006) ..." Wrong citation. Paper by Kowalczuk et al., (2006) said nothing about relationship between aCDOM(350) and DOC. This relationship has been presented for Baltic Sea surface waters (not Gulf of Finland) in paper by Kowalczuk et al., (2010) (Oceanologia, 52(3), 431-471). Remove citation to Kowalczuk et al., 2006.

Responses: The authors thank for the very specific comment, the right study site and

right reference literature were incorporated in the revised manuscript.

Page 7 Lines 131 – 134 " According to Fig.1, the proposed hypothesis suggests that the main source of ...." Repetition. Please try to keep different thread together, do not repeat things that you have said before.

Responses: The authors thank for the valuable comments, these repetitions were avoided in the revised manuscript.

Materials and Methods Page 9 Line 178 " ... converted to in situ salinity units (PSU) in the laboratory. "

The salinity in practical salinity scale has no units – it's a ratio of water electrical conductivity measured at given temperature and pressure to ratio of electrical conductivity of artificial sea water measure at standard temperature and pressure. This phrase shall be written as follow: ... converted to in situ salinity, expressed in practical salinity scale (PSU), in the laboratory.

Responses: The authors really appreciated the valuable suggestion, which has been adopted in the revised manuscript.

Page 9 Line 190

"Chlorophyll-a (Chl-a) was extracted and concentration was measured using a Shimadzu UV-2050PC spectrophotometer (Song et al., 2013)." Detailed method of spectroscopic measurements of chlorophyll a concentration shall be given, or at least a proper reference to equation that converts measure absorbance of pigments extract to chlorophyll a concentration shall be cited. Song et al., are not authors of this method, it has been proposed first by Strickland and Parsons, 1972. Responses: The authors thank for the instructive comments, and the proper citation was added in the revised manuscript.

Results and discussion The whole section shall be rewritten to two sections: Results - where Authors presents their own results, and Discussion – where Authors give interpretation of their results.

Responses: The authors for the instructive comments, as aforementioned, this section was divided into Results and Discussion sections in the revised manuscript.

Page 11 Line 219

Chl-a concentrations ($46.44\pm59.71$ $\mu$g/L) changed from 0.28 to $521.12\mu$g/L, with the-mean of 46.44 $\mu$g/L.

Redundancy – you give the same value of averaged chlorophyll a concentration twice in the same sentence. Correct.

Responses: The authors for the instructive comments, the redundancy was avoided in the revised manuscript by deleting "with the mean of 46.44 $\mu$g/L".

Page 14 Lines 285 – 287

Phytoplankton degradation may contribute relative large portion of CDOM and DOC in these water bodies (Zhang et al., 2010), due to the lower molecular weight, its absorption is different from that derived from terrestrial systems (Helms et al., 2008). Wrong citation again. Helms et al., 2008 neither worked in fresh water bodies nor studied phytoplankton degradation products. They have focused on photobleaching effect on spectral slope and have established a spectral slope ratio as proxy for molecular weight. I do not see any information on spectral slope ratio in this paper – so why do you discuss with Helms et al., 2008. This paper does not present any CDOM absorption spectral slope data at all.

The same wrong citation to Helms et al., (2008) repeated on the same page at line 291.

Responses: The authors thank for the comments. There might a misunderstanding for the reference, here the authors try to say that phytoplankton degradation may change the spectral slope due the change of molecular weight for some components of the

mixture compounds. The wrong citation was removed, and the proper ones were added in the revised manuscript.

Page 14 – 15, Lines 297 - 300

"As suggested by Brezonik et al. (2015) and Cardille et al. (2013), CDOM in the eutrophic waters or those with very short resident time may show seasonal variation due to algal bloom or hydrological variability, while CDOM in some oligotrhopic lakes or those with long resident time may show an opposite pattern." This is a part of discussion, but I do not know which part of results is discussed here. Authors did not spent a lot of time on trophic status of studied lakes. The chlorophyll a is mentioned only in one sentence at the beginning of Results section.

Responses: The authors really appreciated this comments. This sentence was removed since it did not have a strong link with the current study, and we did not pay much attention to the impact of eutrophication on CDOM absorption characteristics.

Page 15 Line 318 " ... were found and less colored portion of DOC was presented in waters in semi-aridto arid regions ... "

I did not found any data on aCDOM(ïAЁŻЁĞn)/DOC relationship in this paper, neither in the text, tables nor figures. What Authors refer to?

Responses: The authors thank for this comments, and sorry for the misleading. In the first submitted version, the relationship between DOC and CDOM were analyzed based on the SUVA254 classification, which has connection with aCDOM(ïAЁŻЁĞn)/DOC relationship, this part was not full removed from the previous version, that caused the misunderstanding. We remove this sentence in the revised manuscript, thanks again for the valuable comments.

Page 16 Line 339 " ... which is consistent with the findings from Helm et al. (2008) ..." Wrong citation again. There is no single line in paper by Helms et al., (2008) on DOC vs. aCDOM(l) relationship.

[Figure]

Response: The authors thank for the comment, there might a misunderstanding for the expression. Here the author did not state the relationship between DOC and CDOM, rather, we tried to say that CDOM in head waters tend to have high molecular weights, thus lower spectral slope values, which has nothing to do with the relationship between CDOM-DOC. To avoid misunderstanding, we rephrased this sentence in the revised manuscript.

Page 19 Line 397 " ... ice and snow cover shielded out most of the solar radiation that might cause a series of biochemical process for CDOM contained in water ..." What specific processes Authors refer to? Citation need to support this statements, otherwise I suggest to delete it.

Response: Thanks for the comment, this sentence was deleted in the revised manuscript.

Page 20 Line 428 "This has important implication for remote sensing of DOC through the CDOM absorption as a bridge (Zhu et al., 2014; Kuster et al., 2015; Brezonik et al., 2015)." What kind of bridge CDOM absorption is ?

Response: Thanks for the comment, we rephrased this sentence in the revised manuscript to make it clear. Here, the authors try to say that CDOM is a optically active constituent that can be remotely sensed, but not DOC. Remote sensing of DOC is based on the relationship between DOC-CDOM, thus CDOM absorption is a bridge for DOC estimate through remotely sensed data.

Page 23 Line 491 "Most of the paired data sitting close to the regression line except some scattered ones." Very bizarre sentence that contains no useful information. Delete it.

Response: Thanks for the comment, this sentence was deleted in the revised manuscript.

Conclusion Delete first two sentences that refer to remote sensing. This paper is

about DOC vs. aCDOM(ïAËŻEǦn) relationships in different water bodies not about remote sensing algorithms.

Response: The authors really thank for the very valuable comments, the first two sentences were removed as suggested.

Page 24 Lines 514 – 516 The slope values of saline lakes and urban waters were close to unity, slope values of river water were highest (âĹij 3.1), and slope values of other water types were in between.Repetition of results – consider to delete.

Response: The authors really thank for the very valuable comments, theserepetitive statements were removed in the revised manuscript.

Acknowledgements "Last but not the least, the authors 534 would like to thank the editor and two anonymous referees ....." Has this manuscript been submitted to other journal and reviewed before current review?

Response: thanks for the comment, yes, this manuscript was submitted to HESS in 2016, and the handling editor (Professor Stamm) suggested to resubmit to HESS, thus the previous acknowledgements were kept.

Figure 3 and 5, 8, 9 Y axis legend on figure 3, 5, 8, 9 Is: aCDOM275 (m-1), should be aCDOM(275) [m-1] – please correct accordingly in all specïfied figures.

Response: The authors really thank for the very valuable comments, Figure 3, 5, 8, and 9 were reproduced with the suggested labels.

Figure 4 Add information to legend – what CDOM absorption coefficient, aCDOM(ïAËŻEǦn) is presented on 3 panel of Figure 4.

Response: The authors really thank for the very helpful comments, CDOM absorption coefficient wavelength of three panels in Figure 4 were added in the revised manuscript.

Figure 6 The same remark as for figures 3, 5, 8, 9 – correct Y axis legend to aC-

DOM(440) [m-1] Figure 6.

Response: The authors really thank for the very valuable comments, the Y axis legend for Figure 6 wascorrected in the revised manuscript.

Figure 7 legend the ratio shall be defined as aCDOM(250)/aCDOM(365) - not aCDOM(250/365). Panel a Y axis SUVA(254) dimension is [m2 g-1].

Response: The authors really thank for the very valuable comments, all your kind suggestions were incorporated in the revised manuscript.

Figure 9 Scales on panel c graph shall be expressed in decimal logarithms log-log. The regression shall be fitted to power function–so it will be linear in log-logscale. See examples in paper by Kowalczuk et al., (2010) (Oceanologia, 52(3), 431-471).

Response: Thanks for the valuable comments, panel c of figure 9 was reproduced as suggested.

Table 2 Add units to DOC and aCDOM(440) as in Table 1.

Response: Thanks for the suggestion, units for DOC and aCDOM(440) were added in Table 2.

---

## Author Comment (AC5) · 24 Jun 2017

I might wrong, I found this comments are same the one by Professor Kowalczuk, thus no further responses were provided. Thanks!

---

## Editor Decision (ED1)

Editor comments on HESS 2017-179

29 Aug 2017

Dear Dr. Song

Thanks for your revision of the manuscript. I think you have addressed most of the concerns and improved the manuscript substantially. Before I can accept if for publication though, some minor aspects need to be rectified. I first list issues related to your answers, subsequently I add those points that refer directly to the manuscript.

Comments on your response:

L. 1562: I did not find such a map. Could you include it into the Supplementary Information? Is there a reason not to provide the coordinates in a table as well?

L. 1604: I did not find these hypotheses. Can you clarify?

L. 1624: Perhaps I missed these metrics in the revised version. If not, please add them.

Comments on the manuscript:

L. 23: Do you mean stronger in a statistical sense or a steeper slope of the regression?

L. 25: Replace *tracer* by *measure*.

L. 61: Insert *natural* before *external*.

L. 68: *Originating* instead of *originates*.

L. 69: Insert *sources* after *anthropogenic*.

L. 74 – 75: Mention also land use and travel times.

L. 86: Changes over what?

L. 98: What do you mean by *circumstances*?

L. 101: … *regions are generally exposed to* …

L: 104: … *of* CDOM, …

L. 104: … are much *more* …

L. 105: Replace *substance* by *factors*.

L. 109: Sentence is not clear.

L. 121: Why *The* significant relationship? Should it not be *A* significant ….

L. 132: Replace *However* by *In addition* or a similar wording.

L. 186: Move *in the laboratory* to the line above after the parenthesis.

L. 232: Replace *changed* by *ranged*.

L: 242: Reduce the number of digits.

L. 253 – 255: This is a weird sentence, reword.

L. 287: Skip *spectra*.

L. 290: What do you mean by stable?

L. 292: *participation* sounds weird in this context, reword.

L: 295 – 297: What is the linkage between the slope and the trophic state of the water bodies?

L. 316 – 320: Reword these sentences; they are linguistically very repetitive (i.e. three times *salines lakes*).

L. 327: Photo-bleaching is an interpretation here, not an empirical result. Add *probably* to the sentence to make this clear.

L. 371: The reason for this strong (meaning here?) is not evident. Clarify.

L. 379: Where can one see that? Why is it a case in-between? Clarify.

L. 387 – 397: This paragraph is poorly structured. Its logic is not obvious since it combines different aspects. Rephrase.

L. 431: What does *its* refer to?

L. 436 – 438: Where can one see this? Perhaps include a table in the Supplementary Material that compares the different slopes.

L. 479 – 483: I suggest that you combine all data in one single figure displaying the data of the different clusters by different colours. By doing so, it should get evident that the data fall into separate groups. You may consider using an in-set to account for the different ranges covered by the different M classes.

L. 541 – 542: This is repetitive.

L. 557: Why *Similarly*?

L. 590 – 602: This is repetitive.

L. 608 – 617: This is repetitive, shorten.

L. 615 – 618: This sentence is weird, reword.

L. 629: What is close, what is scattered? Without any quantitative metric it is a trivial statement that holds true basically for every regression.

L. 653: Insert *probably* after *values*.

L. 654 – 665: This is repetitive, shorten or skip.

L. 683: Acknowledge also the reviewers of the previous version.

Fig. 1: CDOM sources are a subset of DOC sources. This should be made clear in this figure.

Fig. 2: The colours of the different regions and the lines indicating boundaries between them seem not to match. Please clarify in the figure or text.

Supplementary material:

Tables S1 – S3: Replace *Water types* by *Water body type*.

Table S1: There is no specific date for the sampling at the Three Gorges sites. Why?

Table S3: What is the meaning of *Water number*?

Sincerely

Christian Stamm, Editor HESS

---

## Author Response (AR2)

Dear Professor C. Stamm,

Thank you so much for all your efforts in improving our manuscript. We corrected the manuscript according to your kind suggestions or comments, and please find the responses to your comments one by one attached below.

We reproduced Figure 1 and Figure 2 according to your suggestions, and also Figure 8 was reproduced with all data were presented in one diagram. In the process, we find that seven sampling points were missing in Figure 9, which might have regarded as outliers in the previous version of the manuscript. For consistency, we added these points in Figure 9, and regressions were done with each of the sub plot in Figure 9. Accordingly, the main text was changed accordingly. Some of the paragraphs or sentences, which you thought that they were repetitive, were removed. Also, two more tables were added in the supplementary materials, with one deals with the sampling stations for river and stream waters, and one deals with the comparison of regression model parameters from both Figure 3 and Figure 6.

We went through the manuscript again, and found some minor problems that also have been fixed. By the way, for my carelessness, most of the change tracks were not visible now due to my accidental operation on the revised manuscript. We hope this manuscript is acceptable now for potential publication with HESS. We really thank you and these reviewers from both the current and the previous versions of the manuscript, these helpful comments and suggestions definitely strengthened our manuscript.

Sincerely,

Kaishan Song

Editor comments on HESS 2017－179
Aug 2017

Dear Dr. Song
 Thanks for your revision of the manuscript. I think you have addressed most of the concerns and improved the manuscript substantially. Before I can accept if for publication though, some minor aspects need to be rectified. I first list issues related to your answers, subsequently I add those points that refer directly to the manuscript.

Comments on your response:

L. 1562: I did not find such a map. Could you include it into the Supplementary

Information? Is there a reason not to provide the coordinates in a table as well?

**Responses:** thanks for your concern, we provided the information in the
supplementary material, please check it out in the revised manuscript. Since the
sampling stations were already presented in Figure 1, thus we provide these
coordinates in a table added in the supplementary material (Table S4).

L. 1604: I did not find these hypotheses. Can you clarify?
**Responses:** thanks for your concerns, the authors added a sentence, combined with
Figure 1, are the hypotheses for this manuscript. Actually, Figure 1 is the main idea for
this manuscript, which combined with the sentence added in the revised manuscript
illustrates the hypotheses for this manuscript.

L. 1624: Perhaps I missed these metrics in the revised version. If not, please add the
m.
**Responses:** thanks for your concerns, we add p-values in these figures. Perhaps I might
have misunderstood what the reviewer's meaning?

Comments on the manuscript:
L. 23: Do you mean stronger in a statistical sense or a steeper slope of the regression?
**Responses:** thanks for your comment, here we mean stronger in a statistical sense. To
avoid ambiguous statement, we replaced "stronger" with "closer" in the revised
manuscript.

L. 25: Replace tracer by measure.
**Responses:** we thank you for the suggestion, and the authors did the revision as
suggested.

L. 61: Insert natural before external.
**Responses:** the authors really thank for the suggestion, which has been incorporated
in the revised manuscript.

L. 68: Originating instead of originates.
**Responses:** The suggestion has been incorporated in the revised manuscript.

L. 69: Insert sources after anthropogenic.
**Responses:** the authors really thank for the suggestion, which has been incorporated
in the revised manuscript.

L. 74 – 75: Mention also land use and travel times.
**Responses:** the authors really thank for the suggestion, which has been incorporated
in the revised manuscript.

L. 86: Changes over what?
**Responses:** the authors really thank for your concern, here the word "changes" might not a good choice, and we replace it with "variation". Here, we try to say that the
variation of CDOM molecular weight can be tracked using M values.
L. 98: What do you mean by circumstances?
**Responses:** the authors thank for your comment. We rephrased this sentence by
removing "Under the circumstance" in the revised manuscript.
L. 101: … regions are generally exposed to …
**Responses:** the kind suggestion was incorporated in the revised manuscript.
L: 104: … of CDOM, …
**Responses:** the kind suggestion was incorporated in the revised manuscript.
L. 104: … are much more …
**Responses:** the kind suggestion was incorporated in the revised manuscript.
L. 105: Replace substance by factors.
**Responses:** the kind suggestion was incorporated in the revised manuscript.
L. 109: Sentence is not clear.
**Responses:** the authors really thank for the comment. This sentence was rephrased to
achieve a clear statement, and please check it out in the revised manuscript.
L. 121: Why The significant relationship? Should it not be A significant ….
**Responses:** thanks for the comment, "A significant" is appropriate, and it was
incorporated in the revised manuscript.
L. 132: Replace However by In addition or a similar wording.
**Responses:** the kind suggestion was incorporated in the revised manuscript.
L. 186: Move in the laboratory to the line above after the parenthesis.
**Responses:** thanks, your kind suggestion was incorporated in the revised manuscript.
L. 232: Replace changed by ranged.
**Responses:** your kind suggestion was incorporated in the revised manuscript.
L: 242: Reduce the number of digits.
**Responses:** the suggestion was accepted and please check it out in the revised
manuscript.
L. 253 – 255: This is a weird sentence, reword.
**Responses:** thanks for your comment, and this sentence was rephrased in the revised
manuscript. Here, the author try to say that DOC concentration in the ice melting
waters demonstrated the lowest value in all types of waters.

L. 287: Skip spectra.

**Responses:** your kind suggestion was incorporated in the revised manuscript.

L. 290: What do you mean by stable?

**Responses:** thanks for your concern, we use "stable" to express that the relationship between DOC and $a_{CDOM}(275)$, $a_{CDOM}(295)$ is close, and reliable regression model can be established through linking DOC to CDOM absorption at 275 and 295 nm. Thus, the word "stable" was kept in the manuscript.

L. 292: participation sounds weird in this context, reword.

**Responses:** thanks for your comment, we replaced "participation" with "inclusion" in the revised manuscript.

L: 295 – 297: What is the linkage between the slope and the trophic state of the water bodies?

**Responses:** the authors thank for your comment. The regression model slope values have connections with the CDOM sources, for instance, waters from head rivers or streams generally exhibit higher absorbing efficiency or specific absorption, thus the regression slope is higher, while saline water shows the inverse trend. As we know that the trophic state is closely associated with internal CDOM source originated from algae, while CDOM originated from algae usually contains small size molecular organic compound, thus the regression slope between CDOM and DOC is lower than that from river waters.

L. 316 – 320: Reword these sentences; they are linguistically very repetitive (i.e. three times saline lakes).

**Responses:** the authors really thank for the valuable comment. We rephrased this sentence, and these repetitive parts were removed, thanks again for the suggestion.

L. 327: Photobleaching is an interpretation here, not an empirical result. Add probably to the sentence to make this clear.

**Responses:** the authors really thank for the valuable comment. We removed this part to the Discussion section (4.2), and rephrased in the revised manuscript as well.

L. 371: The reason for this strong (meaning here?) is not evident. Clarify.

**Responses:** the authors thank for the valuable comment, the coefficient of determination of the regression model was added in the sentence to support the statement.

L. 379: Where can one see that? Why is it a case in‐between? Clarify.

**Responses:** thanks for the comment, please see the response for L. 371.

L. 387 – 397: This paragraph is poorly structured. Its logic is not obvious since it comb ines different aspects. Rephrase.

**Responses:** the authors really thank for the valuable comment, we reorganized this paragraph, and tried the best to make it logically appropriate.

L. 431: What does it refer to?

**Responses:** the authors thank for the concern, it here represents CDOM in this context.

L. 436 – 438: Where can one see this? Perhaps include a table in the Supplementary

Material that compares the different slopes.

**Responses:** the authors really thank for the valuable comments, a table was produced and added in the supplementary material.

L. 479 – 483: I suggest that you combine all data in one single figure displaying the data of the different clusters by different colours. By doing so, it should get evident that the data fall into separate groups. You may consider using an inset to account for the different ranges covered by the different M classes.

**Responses:** the authors really thank for the valuable comments, we combined all data in one single figure as suggested. Also, we adjusted the hierarchical cluster a little bit since M < 8.5 and M<9 (M<25 and M<25.6) won't make big difference, thus the regression models and sample numbers were slightly different from the previous ones.

Further, by combing all the data in single figure, we found that 7 samples were not included in Figure 9 (these in the circle) in the previous version. It might have been regarded as outliers in the data analysis process, to keep data consistency, we added these samples in Figure 9 in the revised manuscript, and correspondingly, the main text was changed accordingly.

[Figure]

L. 541 – 542: This is repetitive.

**Responses:** the authors really thank for the valuable comments, these repetitive parts were removed in the revised manuscript.

L. 557: Why Similarly?

**Responses:** the authors thank for the concern, this sentence was rephrased to avoid ambiguity.
L. 590 – 602: This is repetitive.
**Responses:** the authors thank for your comments, this paragraph is removed.
L. 608 – 617: This is repetitive, shorten.
**Responses:** the authors really thank for your comments, the repetitive parts were
removed and this paragraph is shortened.
L. 615 – 618: This sentence is weird, reword.
**Responses:** the authors thank for your concern, this sentence is removed.
L. 629: What is close, what is scattered? Without any quantitative metric it is a trivial
statement that holds true basically for every regression.
**Responses:** the authors thank for your concern, this sentence is removed.
L. 653: Insert probably after values.
**Responses:** thanks for the suggestion, it has been incorporated in the revised
manuscript.
L. 654 – 665: This is repetitive, shorten or skip.
**Responses:** the authors thank for your suggestion, this sentence is removed.
L. 683: Acknowledge also the reviewers of the previous version.
**Responses:** the authors thank for your suggestion, we also acknowledged the
reviewers of the previous version of the manuscript.
Fig. 1: CDOM sources are a subset of DOC sources. This should be made clear in this
figure.
**Responses:** the authors really thank for your suggestion, modification was made for
Figure 1, please check it out in the revised manuscript.
Fig. 2: The colours of the different regions and the lines indicating boundaries between
them seem not to match. Please clarify in the figure or text.
**Responses:** the authors really thank for your very valuable suggestion, we reproduced
Figure 2, and different regions in Figure 2 is matching that in the main text now.
Supplementary material:
Tables S1 – S3: Replace Water types by Water body type.
**Responses:** thanks for the suggestion, we did the revision as suggested.
Table S1: There is no specific date for the sampling at the Three Gorges sites. Why?
**Responses:** thanks for your concern, we added the starting date for the sampling at the Three Gorges site in the revised manuscript. The Three Gorges Reservoir is an
elongated water body, thus it took us three days to finish this field survey in a fishing
boat, which was rather slow. The starting date was supplied in the supporting Table 1.
Table S3: What is the meaning of Water number?
**Responses:** thanks for your concern, here the authors try to say how many water
bodies were sampled in each city, we changed "water number" to "Water body" in the
revised manuscript.
Sincerely
Christian Stamm, Editor HESS

[revised manuscript text omitted]

---

## Author Response (AR3)

**Editor Decision: Publish subject to minor revisions (further review by Editor)** (04 Sep

2017) by Christian Stamm

Comments to the Author:

Dear Dr. Song

Thanks for your Revision. There a just two technical details:

My comment to L. 1624: I was probably not clear want I exptected. You mentioned in your first response that you provide the regressions for Fig. 3 etc. without points that can be considered outliers. I had these metrics in mind. It would be nice to add a table in the SI listing These regressions (including R2, p values) and refering to that table in the figure captions.

**Response:** the authors thank you for the concern and detailed suggestion, we did the regressions without these high DOC values, which is preferentially regarded as outliers, and all the regression metrics (i.e., coefficient of determination ($R^2$), slope, intercept and p-value) were provided in Table S6. Further, the main text was changed correspondingly with respect to the supplementary Table S6. We really thank you for the kind suggestion.

L. 328 in the latest Version: I suggest to correct to "... CDOM essentially originates from ...".

**Response:** we thank you for the correction of the grammatical problem, and your kind suggestion was incorporated in the revised manuscript.

Upon these modifications I can accept the manuscript.

Sincerely

Christian Stamm

[revised manuscript text omitted]